# BOND:
## ALIGNING LLMS WITH BEST-OF-N DISTILLATION

**Pier Giuseppe Sessa,[†] Robert Dadashi, Léonard Hussenot, Johan Ferret, Nino Vieillard**
**Alexandre Ramé, Bobak Shahriari, Sarah Perrin, Abram L. Friesen, Geoffrey Cideron**
**Sertan Girgin, Piotr Stanczyk, Andrea Michi, Danila Sinopalnikov, Sabela Ramos**
**Amélie Héliou, Aliaksei Severyn, Matt Hoffman, Nikola Momchev, Olivier Bachem**

Google DeepMind

### ABSTRACT

Reinforcement learning from human feedback (RLHF) is a key driver of quality and safety in state-of-the-art large language models. Yet, a surprisingly simple and strong inference-time strategy is Best-of-N sampling that selects the best generation among $N$ candidates. In this paper, we propose Best-of-N Distillation (BOND), a novel RLHF algorithm that seeks to emulate Best-of-N but without its significant computational overhead at inference time. Specifically, BOND is a distribution matching algorithm that forces the distribution of generations from the policy to get closer to the Best-of-N distribution. We use the Jeffreys divergence (a linear combination of forward and backward KL) to balance between mode-covering and mode-seeking behavior, and derive an iterative formulation that utilizes a moving anchor for efficiency. We demonstrate the effectiveness of our approach and several design choices through experiments on abstractive summarization and Gemma models.

## 1 INTRODUCTION

State-of-the-art large language models (LLMs) such as Gemini (Gemini Team, 2023; Reid et al., 2024) and GPT-4 (OpenAI, 2023) are generally trained in three stages. First, LLMs are pre-trained on large corpora of knowledge using next-token prediction (Radford et al., 2018; 2019). Second, the pre-trained models are fine-tuned to follow instructions via supervised fine-tuning (SFT) (Raffel et al., 2020; Wei et al., 2022). Lastly, reinforcement learning from human feedback (RLHF) (Christiano et al., 2017; Ziegler et al., 2019; Stiennon et al., 2020) is used to further increase the quality of generations. The RLHF step generally consists of learning a reward model (RM) (Ouyang et al., 2022) on human preferences and then optimizing the LLM to maximize predicted rewards using reinforcement learning algorithms.

**RLHF algorithms and their challenges.** Fine-tuning LLMs with reinforcement learning (RL) is challenging (Casper et al., 2023), notably since it can cause *forgetting* (French, 1992) of pre-trained knowledge, and since loopholes in the RM (Clark & Amodei, 2016; Pan et al., 2022) can cause *reward hacking* (Askell et al., 2021; Skalse et al., 2022). The standard strategy is to use policy-gradient methods (Williams, 1992) with KL regularization towards the SFT policy. Those RL algorithms seek Pareto-optimal policies with high reward at low KL, to preserve the general capabilities of the original model and tackle the misalignment (Ngo et al., 2022) concerns.

**Best-of-N sampling.** In practice, a surprisingly simple inference-time approach is often used to improve the quality of generations: Best-of-N sampling (Stiennon et al., 2020). It consists of drawing $N$ candidate generations from the reference (typically, supervised fine-tuned) model and selecting the one with the highest reward according to the RM. This strategy empirically achieves excellent reward-KL trade-offs (Nakano et al., 2021; Gao et al., 2023; Touvron et al., 2023) but increases the computational cost by a factor of $N$.

**BOND.** In this paper, we propose BOND (Best-of-N Distillation), a novel RLHF algorithm that *learns* a policy that achieves the strong performance of Best-of-N sampling but, crucially, requires only a

---

[†]Correspondence to: Pier Giuseppe Sessa <`piergs@google.com`>

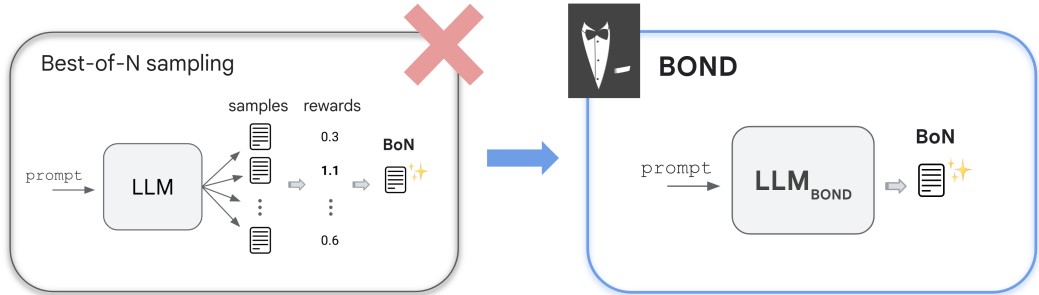

Figure 1: Best-of-N is an *inference-time* strategy that selects the best generation among $N$ candidates from a reference LLM policy, improving quality at the cost of a large computational (need to sample and score $N$ times from the model). In contrast, the proposed BOND approach aims at obtaining a fine-tuned policy that can directly sample the Best-of-N generation. This would inherit the quality of Best-of-N sampling, while requiring a single sample at inference time. We achieve this by distilling the Best-of-N strategy into the policy via online *distribution matching*.

single sample at inference time, as depicted in Figure 1. Our key idea is to cast the alignment of the policy as a distribution matching problem, where we fine-tune the policy to emulate the Best-of-N distribution. To achieve this, we first derive an analytical expression for the Best-of-N distribution. This allows us to consider and optimize different divergence metrics. We first show how to minimize the *forward* KL divergence using samples from the Best-of-N strategy, leading to a standard imitation learning setup with a mode covering behavior. We also show how to minimize the *backward* KL, leading to a new form of quantile-based advantage, which does not depend on the reward scale, and corresponds to a mode seeking behavior. Then, we propose to minimize a linear combination of forward and backward KL, also known as *Jeffreys divergence*, which retains the best of both approaches. Furthermore, to optimize performance while keeping a reduced sample-complexity, we propose an *iterative* BOND approach which consists of iteratively distilling the Best-of-N of a moving anchor policy. Finally, based on the aforementioned ideas, we propose J-BOND (J for Jeffreys), a novel, stable, efficient and practical RLHF algorithm to align LLMs.

**Experiments.** We first demonstrate the effectiveness of BOND and of our design choices on the abstractive summarization XSum (Narayan et al., 2018) task. Then, in Section 6, we apply J-BOND to align Gemma (Gemma Team, 2024) policies. J-BOND does not require committing to a specific regularization strength, but it continuously improves the reward displaying a stable optimization and a better reward/KL trade-off compared to standard RL algorithms. This translates into a higher quality and improved scores on popular real-world benchmarks.

## 2  PROBLEM SETUP

We consider a LLM based on the transformer (Vaswani et al., 2017) architecture, defining a policy $\pi(x, \cdot)$ by auto-regressively generating token sequences $y$ from the prompt $x$. Given a pre-trained and typically supervised fine-tuned reference policy $\pi_{\text{ref}}$, we seek to further align it to human preferences. To achieve this, throughout the rest of the paper we assume access to a reward model (RM) which we denote as $r(\cdot)$, trained to reflect human preferences.

**Standard RLHF.** Most RL algorithms optimize a linear combination of the expected reward and a KL divergence between the current and reference policy:

$$\pi_{\text{RL}} = \text{argmax}_\pi \, \mathbb{E}_\pi[r(y)] - \beta_{\text{RL}} \cdot \text{KL}(\pi \, || \, \pi_{\text{ref}}), \quad (1)$$

with regularization strength $\beta_{\text{RL}} \geq 0$. This KL regularization forces the policy to remain close to its initialization $\pi_{\text{ref}}$ (Geist et al., 2019; Lazaridou et al., 2020), reducing forgetting (French, 1992) and reward hacking (Skalse et al., 2022). Equation (1) is usually optimized with online algorithms, as they perform better than their offline counterparts (Tang et al., 2024). Moreover, simple methods have demonstrated the best results, e.g., REINFORCE (Williams, 1992) with a sampled baseline for variance reduction (Li et al., 2023; Ahmadian et al., 2024) outperform PPO (Schulman et al., 2017).

**Best-of-N.** A complementary alignment strategy is Best-of-N, which is an inference-time strategy that involves sampling multiple times from $\pi_{\text{ref}}$ and selecting the generation with highest reward

according to the RM $r$. In contrast to RLHF strategies, Best-of-$N$ does not fine-tune the weights of the LLM, but instead modifies the inference procedure. Best-of-$N$ was empirically shown to be efficient (Touvron et al., 2023) when looking at reward/KL trade-offs, and comes with theoretical guarantees (Qiping Yang et al., 2024) in terms of Pareto-optimality. Unfortunately, Best-of-$N$ comes at a significantly higher inference cost which increases linearly with $N$, since producing $N$ generations is (in general) $N$ times more costly than sampling a single one.

Motivated by the above considerations, we propose a novel alignment method which we name BOND for Best-of-$N$ Distillation. The goal of BOND is to *distill the Best-of-$N$ strategy into the policy*. This allows the policy to reach the strong performance of Best-of-$N$ sampling, while requiring only *a single sample* at inference time. We outline our overall approach in the next section.

## 3 THE BOND APPROACH

We formulate the BOND approach in two main steps. First, we derive an analytical expression for the Best-of-$N$ distribution (Section 3.1). Second, using the derived expression, we phrase the problem as a *distribution matching* problem (Section 3.2), i.e., we want to steer the policy closer to the Best-of-$N$ distribution. In Section 3.3, we draw insightful connections between BOND and standard RLHF.

### 3.1 THE BEST-OF-$N$ DISTRIBUTION

In this section, we derive the exact analytical distribution of Best-of-$N$ sampling and study its properties. For simplicity, we drop the context $x$ from all notation without loss of generality and assume that the reward $r(y)$ induces a strict ordering on all generations $y$[1]. We can affirm the following main theorem (proof in Appendix A.1).

**Theorem 1.** *For any generation $y$, let*

$$p_<(y) = \mathbb{P}_{y' \sim \pi_{ref}}[r(y') < r(y)] \tag{2}$$

*denote the probability that a random generation $y'$ from $\pi_{ref}$ is strictly worse than $y$ and let*

$$p_\le(y) = \mathbb{P}_{y' \sim \pi_{ref}}[r(y') \le r(y)], \tag{3}$$

*the probability that $y'$ is not better than $y$ (thus including the equality case). Then, the probability that $y$ is the output of Best-of-$N$ sampling is given by*

$$\pi_{BoN}(y) = \pi_{ref}(y) \times \underbrace{p_\le(y)^{N-1}}_{(A)} \times \underbrace{\sum_{i=1}^{N}\left[\frac{p_<(y)}{p_\le(y)}\right]^{i-1}}_{(B)}. \tag{4}$$

**Interpretation.** Theorem 1 provides an intuitive explanation on the behavior of Best-of-$N$ sampling: it essentially reweights the original sampling distribution $\pi_{\text{ref}}$, by the multiplicative terms (A) and (B).

The term (A) corresponds to a penalty exponential in $N$ based on the fraction of generations (for the same prompt) that are worse or equal to the considered generation $y$. Intuitively, this ensures that we sample exponentially less from bad generations when increasing $N$.

The term (B) is an additional correction factor due to the potential of collisions among generations. Importantly, it is at most linear in $N$ as it is always bounded within $[1, N]$:

$$1 \le 1 + \sum_{i=2}^{N}\left[\frac{p_<(y)}{p_\le(y)}\right]^{i-1} = \sum_{i=1}^{N}\left[\frac{p_<(y)}{p_\le(y)}\right]^{i-1} \le \sum_{i=1}^{N} 1 \le N. \tag{5}$$

It achieves its minimum at 1 for the worst generation $y_-$ since we have $p_<(y_-) = 0$ by definition. This is not surprising, as we need to sample $y_-$ exactly $N$ times in a row and which corresponds to $\pi_{\text{BoN}}(y_-) = \pi_{\text{ref}}(y_-)^N$ (note that $p_\le(y_-) = \pi_{\text{ref}}(y_-)$). In contrast, if the likelihood of individual generations $y$ are low and such generations are good, then $p_<(y)$ is almost $p_\le(y)$ and term (b) is close to $N$. Intuitively, this corresponds to the case where sampling a generation $y$ multiple times is unlikely. In the extreme case when $\pi_{\text{ref}}$ is a continuous distribution, term (B) is constant and equal to $N$ (see Appendix A.2).

---

[1]To distinguish between generations with the same reward, ties can be broken by any arbitrary strict ordering.

## 3.2 THE BOND OBJECTIVE

The analytical characterization of the Best-of-$N$ distribution allows us to formulate BOND as a distribution matching problem. That is, we want to solve the objective:

$$\pi_{\text{BOND}} = \arg \min_{\pi \in \Pi} D(\pi \, \| \, \pi_{\text{BoN}}), \tag{6}$$

where $D(\cdot \, \| \, \cdot)$ is a divergence metric steering the training policy $\pi$ towards $\pi_{\text{BoN}}$. For this, a toolbox of possible divergences exist in the literature including, e.g., forward and backward KL (Kullback, 1959). Moreover, we can employ existing distribution matching techniques to estimate $D$ from online and offline samples. We defer the choice of divergences and resulting BOND algorithms to Section 4.

## 3.3 CONNECTION WITH STANDARD RLHF

In this section, we draw important connections between the two seemingly different objectives of standard RLHF (Equation (1)) and BOND (Equation (6)).

It is well known (see, e.g., Vieillard et al., 2020; Rafailov et al., 2023) that the policy maximizing the RLHF objective from Equation (1) is:

$$\pi_{\text{RL}}(y) \propto \pi_{\text{ref}}(y) \exp\left( \frac{1}{\beta_{\text{RL}}} \, r(y) \right). \tag{7}$$

From the derived expression of $\pi_{\text{BoN}}$ in Theorem 1, we see that the Best-of-$N$ sampling distribution coincides with the optimal solution of standard RLHF when using the following specific BOND reward:

$$r_{\text{BOND}}(y) = \underbrace{\log p_{\leq}(y)}_{(\text{A})} + \underbrace{\frac{1}{N-1} \log \sum_{i=1}^{N} \left[ \frac{p_{<}(y)}{p_{\leq}(y)} \right]^{i-1}}_{(\text{B})}, \tag{8}$$

and the specific regularization strength $\beta_{\text{BOND}} = \frac{1}{N-1}$. The term (B) corresponds to the correction factor in Theorem 1, which is bounded in $\left[ 0, \frac{\log N}{N-1} \right]$ for all generations $y$. Instead term (A) lies in $(-\infty, 0]$. This provides two interesting insights for Best-of-$N$ sampling:

1. Best-of-$N$ sampling corresponds to the solution of a standard KL-regularized RLHF problem where the choice of $N$ determines the level of KL regularization.

2. Best-of-$N$ sampling corresponds to optimizing the expected log reward quantile, i.e., the log likelihood that the generation has larger reward than a random sample from the reference distribution. Interestingly, due to the concavity of the logarithm, $r_{\text{BOND}}(y)$ strongly encourages the model to avoid bad generations rather than encouraging to generate good ones. Moreover, $r_{\text{BOND}}(y)$ is *invariant to monotone transformations of the reward* $r(\cdot)$, since it depends only on the rank among the generations. We conjecture that both these features make the BOND reward $r_{\text{BOND}}(y)$ more robust to reward hacking compared to standard RLHF.

The connection to RLHF also inspires the proposed approach in this manuscript: if we can compute the BOND reward or equivalently the Best-of-$N$ distribution $\pi_{\text{BoN}}$, then we can steer the policy towards Best-of-$N$ via distribution matching. In the next section we explore different algorithms to tackle the main underlying challenges.

## 4 BOND CHALLENGES AND ALGORITHMS

Implementing the BOND approach induces the three following challenges: *(1)* how to estimate the reward quantiles, *(2)* which is the appropriate divergence metric to use, and *(3)* how to choose the hyperparameter $N$ representing the number of sampled generations in Best-of-$N$. We discuss and address these challenges in the next three subsections.

## 4.1 MONTE-CARLO QUANTILE ESTIMATION

One key difficulty in estimating the $\pi_{\text{BoN}}$ distribution is that we need to estimate the quantile

$$p_{\leq}(y) = \mathbb{P}_{y' \sim \pi_{\text{ref}}}[r(y') \leq r(y)], \tag{9}$$

of a given generation $y$. The quantile $p_\leq(y)$ measures the quality of $y$ compared to generations from $\pi_{\mathrm{ref}}$ when conditioned on the same prompt (recall that we have suppressed the conditioning on $x$ in our notation). A very simple but effective quantile estimation method is *Monte-Carlo sampling*, sampling $k$ generations from $\pi_{\mathrm{ref}}$ and obtaining the following empirical estimate:

$$\hat{p}_\leq(y) = \frac{1}{k} \sum_{i=1}^k \mathbb{I}\{r(y_i) \leq r(y)\}. \tag{10}$$

We found this to be a very effective in our experiments, even with a limited number of samples. In principle, though, one could also use alternative approaches, e.g., training a learned quantile model (as we explore in Appendix B.2).

## 4.2 JEFFREYS DIVERGENCE AS A ROBUST OBJECTIVE

The choice of the divergence metric used in BOND is of crucial importance: different divergences can steer the policy to very different solutions. Here, we propose the *Jeffreys divergence* as a robust distribution matching objective.

The Jeffreys divergence (Jeffreys, 1946) between two distributions is defined as:

$$J_{\mathrm{effreys}}^\beta(p \,\|\, q) := (1 - \beta) \cdot \underbrace{\mathrm{KL}(q \,\|\, p)}_{\text{Forward KL}} + \beta \cdot \underbrace{\mathrm{KL}(p \,\|\, q)}_{\text{Backward KL}}. \tag{11}$$

The (generalized) Jeffreys divergence is a weighted average (with weight $\beta \in [0, 1]$) between the forward and backward KL divergence. Notably, when fine-tuning policy $p$, the forward $\mathrm{KL}(q \,\|\, p)$ encourages that generations likely under $q$ are also likely under $p$, thus encouraging a *mode-covering* behavior. Instead, the reverse $\mathrm{KL}(p \,\|\, q)$ is well-known to have a *mode-seeking* effect, steering policy $p$ to produce generations that have a high likelihood according to $q$ (Agarwal et al., 2024). While the forward KL may produce over-spread distributions, the backward KL can lead to policy and entropy collapses. Instead, we empirically show that the Jeffreys divergence inherits the best of both divergences, producing better aligned policies.

In the context of BOND, this translates into minimizing the divergence $J_{\mathrm{effreys}}^\beta(\pi \,\|\, \pi_{\mathrm{BoN}})$ which we can estimate using samples from the training policy $\pi$ and reference policy $\pi_{\mathrm{ref}}$ as follows.

**Estimation of the forward KL.** The forward KL defined as

$$\mathrm{KL}(\pi_{\mathrm{BoN}} \,\|\, \pi) = \mathbb{E}_{y \sim \pi_{\mathrm{BoN}}}[\log \pi_{\mathrm{BoN}}(y) - \log \pi(y)] \tag{12}$$

can be estimated directly drawing samples from the $\pi_{\mathrm{BoN}}$ (i.e., sampling $N$ times from $\pi_{\mathrm{ref}}$ and selecting the best one) and can be seen as a supervised fine-tuning loss on the Best-of-N samples:

$$\nabla_\pi \mathrm{KL}(\pi_{\mathrm{BoN}} \,\|\, \pi) = -\mathbb{E}_{y \sim \pi_{\mathrm{BoN}}} \nabla \log \pi(y). \tag{13}$$

**Estimation of the backward KL.** The backward KL defined as

$$\mathrm{KL}(\pi \,\|\, \pi_{\mathrm{BoN}}) = \mathbb{E}_{y \sim \pi}[\log \pi(y) - \log \pi_{\mathrm{BoN}}(y)] \tag{14}$$

can be estimated from the policy samples (note the expectation w.r.t. $\pi$) and their estimated log-likelihood under $\pi_{\mathrm{BoN}}$. In particular, by the analogies drawn in Section 3.3, we show (in Appendix A.3) that its gradient coincides with a policy gradient (e.g., used by REINFORCE (Williams, 1992) in standard RLHF):

$$\nabla_\pi \mathrm{KL}(\pi \,\|\, \pi_{\mathrm{BoN}}) = -(N-1)\mathbb{E}_{y \sim \pi}\big[\nabla_\pi \log \pi(y)\big(r_{\mathrm{BOND}}(y) - \beta_{\mathrm{BOND}}\big(\log \pi(y) - \log \pi_{\mathrm{ref}}(y)\big)\big)\big], \tag{15}$$

with the equivalent reward $r_{\mathrm{BOND}}$ and regularization $\beta_{\mathrm{BOND}}$ defined in Section 3.3. Note that $r_{\mathrm{BOND}}(y)$ depends on the true unknown quantile $p_\leq(y)$ and on the correction factor (B) defined in Equation (8). In practice, we substitute the true quantile by its estimate, while we observed the correction factor does not play a significant role. Thus, we use $r_{\mathrm{BOND}}(y) = \hat{p}_\leq(y)$. Moreover, to reduce the resulting variance, we use a policy gradient baseline (Sutton & Barto, 1998) which we compute as the average return for the other generations in the batch.

Thus, the overall $J_{\mathrm{effreys}}^\beta$ loss is a linear weighted combination between a supervised fine-tuning and a policy gradient loss.

**Experiments.** We consider the abstractive summarization XSum task (Narayan et al., 2018) with $\pi_{\text{ref}}$ being a T5 supervised fine-tuned policy and $r(\cdot)$ being a T5 NLI reward model (Roit et al., 2023). We run BOND with $J_{\text{effreys}}^{\beta}$ objective and $\beta \in \{0, 0.5, 1\}$. We use 16 MC samples (per prompt) to estimate quantiles during training. During eval, we use 32 MC samples to estimate the backward and forward KL divergences between the training policy and the $\pi_{\text{BoN}}$ distribution. We report these in Figure 7 (Appendix B.1) when setting $N = 4, 8$ and 16, respectively. The results confirm our intuition: the Jeffreys divergence ($\beta = 0.5$) allows to minimize both divergences from $\pi_{\text{BoN}}$ (left and middle plot), compared to when solely the backward ($\beta = 1$) or the forward ($\beta = 0$) KL divergence is minimized. In addition, we compute the reward log quantiles (averaged over the eval batches) of the training policy. Interestingly, BOND with $\beta = 0.5$ maximizes the quantiles similarly to the mode-seeking $\beta = 1$ choice, while the mode-covering forward KL ($\beta = 0$) lags behind.

## 4.3 ITERATIVE BOND

Finally, we discuss the choice of the parameter $N$. In practice, choosing $N$ may be difficult for three main reasons: *(1)* As in standard RLHF, $N$ plays the role of regularization (see Section 3.3): a large $N$ improves downstream performance, but if $N$ is too large it will eventually cause reward over optimization (Gao et al., 2023). *(2)* The larger the $N$ the more the estimate of $\pi_{\text{BoN}}$ is sensitive to errors in the estimated quantiles (since $\pi_{\text{BoN}}(y) \propto p_{\leq}(y)^{N-1}$). *(3)* Estimating the forward KL divergence requires sampling from $\pi_{\text{BoN}}$ which is prohibitive for large $N$.

To address the above challenges, we propose the *iterative* BOND approach. The approach is inspired by the fact that Best-of-N sampling from a Best-of-N distribution, coincides with Best-of-N$^2$ sampling from the original distribution. More generally, by informally defining $\text{BoN}(\cdot)$ as an operator that performs Best-of-N sampling from a base distribution, we have:

$$\underbrace{\text{BoN}(\cdots \text{BoN}(\text{BoN}(\pi_{\text{ref}})))}_{M \text{ times}} \equiv \text{BoN}^M(\pi_{\text{ref}}). \tag{16}$$

This suggests the key idea behind iterative BOND: if we know how to distill the Best-of-N distribution (i.e., via BOND), then we can apply BOND recursively (say $M$ times), equivalently to distilling a Best-of-N$^M$ of the initial distribution $\pi_{\text{ref}}$. This allows fixing a small $n$ (i.e., $n = 2$) and running BOND (with $N = n$) in an *iterative fashion*, as an improvement operator. For this, we can introduce an auxiliary *anchor policy* $\pi_{\text{anchor}}$ initialized as $\pi_{\text{ref}}$. We can then run BOND against $\pi_{\text{anchor}}$ (i.e., we can distill the Best-of-n version of $\pi_{\text{anchor}}$) and, after a given number of distillation steps, update $\pi_{\text{anchor}}$ to be the current training policy $\pi_t$. The overall approach is depicted in Figure 2 and summarized in Algorithm 1.

In a nutshell, iterative BOND allows exponential scaling to arbitrary large $N$ (in fact, it does not require setting $N$ in advance) while keeping a reduced sample complexity and a stable optimization. The claim is validated in the results below.

**Experiments.** In Figure 3, we consider the same experimental setup of Section 4.2, fix the BOND objective to $J_{\text{effreys}}^{0.5}$ and run iterative BOND with $n \in \{2, 4\}$ where the moving anchor is updated every 1000 steps. We report the average reward (left plot) and average log quantile (middle plot) obtained during training and compare them to (non-iterative) BOND run with $N \in \{4, 8, 16\}$. As expected, both reward signals saturate early for non-iterative BOND (the smaller the $N$ the earlier the reward saturates), while the iterative BOND approach continuously improves performance (the higher the $n$, the faster). Moreover, in the rightmost plot we plot the obtained log quantiles against the KL

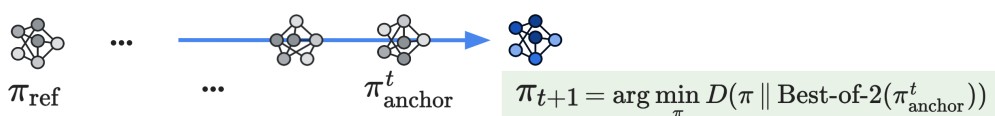

Figure 2: Iterative BOND approach. The policy $\pi_t$ is trained to iteratively distill a Best-of-$N$ (in the figure, $N = 2$) version of a *moving anchor*. This allows to continuously improve the policy performance without requiring to set a (large) $N$ upfront. Moreover, it leads to better training stability and a minimal computational complexity, since a small $n$ is used at each distillation step.

---

**Algorithm 1** Iterative BOND (meta algorithm)

---

**Inputs:** $\pi_{\text{ref}}$, $n \in \mathbb{N}$.
Initialize $\pi_0 = \pi_{\text{ref}}$, $\pi_{\text{anchor}}^0 = \pi_{\text{ref}}$.
**for** $t = 0, \ldots,$ **do**                                                                  */ iterations
$\quad \pi_{t+1} = \arg\min_{\pi \in \Pi} D(\pi \, \| \, \text{Best-of-n}(\pi_{\text{anchor}}^t))$    */ distill the Best-of-n version of $\pi_{\text{anchor}}^t$
$\quad \pi_{\text{anchor}}^t = \pi_t$

---

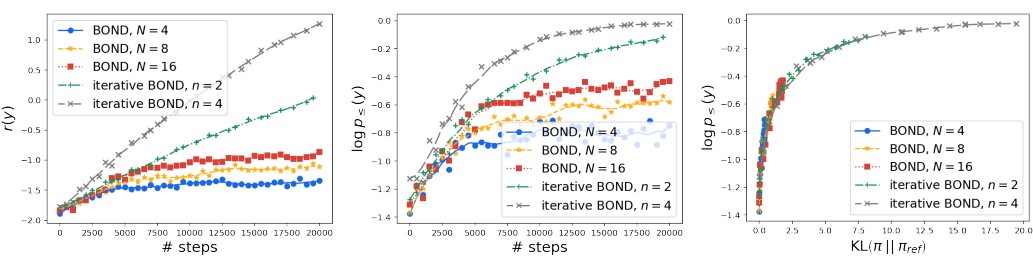

Figure 3: Iterative BOND (with $n = 2$ and $n = 4$) compared to BOND with $N = 4, 8, 16$. Iterative BOND continuously improves rewards (left plot) and log quantiles (middle plot), while they saturate for non-iterative BOND (the smaller the $N$, the sooner). It allows achieving the same reward/KL trade-off (right plot) as non-iterative BOND while keeping a small $n$ and smoothly moving away from $\pi_{\text{ref}}$.

from the reference policy. The plot shows that iterative BOND essentially has the same reward/KL trade-off of the non-iterative BOND runs, but crucially allows keeping a small $n$ and to smoothly but continuously move away from $\pi_{\text{ref}}$.

## 5   THE J-BOND ALGORITHM

In this section we present J-BOND, a concrete and practical BOND algorithm motivated by the results discussed in the previous sections. We describe its main components below, and summarize it in the pseudo-code of Algorithm 2.

J-BOND follows the template of iterative BOND (Algorithm 1) with $n = 2$, i.e., it fine-tunes policy $\pi_t$ to iteratively distill the Best-of-2 version of a moving anchor $\pi_{\text{anchor}}^t$, initialized as $\pi_{\text{ref}}$. The name J-BOND stands for *Jeffreys divergence* BOND because it uses the Jeffreys divergence as distribution matching objective, i.e., it minimizes $J_{\text{effreys}}^{\beta}(\pi \, \| \, \text{Best-of-2}(\pi_{\text{anchor}}^t))$ as defined in Section 4.2.

**Minimal sample complexity.** Compared to the BOND algorithms tested in the previous section, J-BOND has a minimal sample complexity: for each prompt in the batch it generates 1 sample from the policy $\pi_t$ and 2 samples from the anchor $\pi_{\text{anchor}}^t$. While more anchor samples are generally useful for a better divergence estimation (in Section 4 we used 16 MC samples), autoregressive sampling is the main bottleneck of online RLHF and we have thus opted for a practical approach working with a small number of samples.

**Crude divergence estimate based on 2 anchor samples.** The policy and anchor samples are used to obtain a crude estimate of the forward and backward KL components of $J_{\text{effreys}}^{\beta}(\pi \, \| \, \text{Best-of-2}(\pi_{\text{anchor}}^t))$ as described next.

We can minimize the *forward KL* as described in Section 4.2, by doing supervised fine-tuning on the best of the 2 anchor samples. To minimize the *backward KL*, we utilize the policy gradient-style loss of Equation (15) replacing $r_{\text{BOND}}(y)$ with a different reward which we denote as $r_{\text{J-BOND}}(y)$. The reason for this is that when only 2 anchor samples are available, the reward $r_{\text{BOND}}(y) = \log \hat{p}_{\leq}(y)$ would be quite uninformative due to $\hat{p}_{\leq}(y)$ being a very noisy MC estimate. Let $y$ be the policy sample and $\{y_1', y_2'\}$ be the corresponding anchor samples, we instead define $r_{\text{J-BOND}}(y)$ as

$$r_{\text{J-BOND}}(y) = \begin{cases} -\log(16) & \text{if} \quad r(y) < \min\{r(y_1'), r(y_2')\} \\ 0 & \text{otherwise} \end{cases}. \tag{17}$$

---

**Algorithm 2** The J-BOND algorithm

---

**Inputs:** Prompt dataset $\mathcal{D}$, reference policy $\pi_{\text{ref}}$, reward $r(\cdot)$, $\beta, \eta \in [0,1], \gamma \geq 0$.
Initialize policy and anchor $\pi_0 = \pi_{\text{anchor}}^0 = \pi_{\text{ref}}$.
**for** $t = 0, \ldots$ **do**
  Sample batch of prompts $\mathcal{D}_t \subseteq \mathcal{D}$
  For each $x \in \mathcal{D}_t$ generate: 1 policy sample $y \sim \pi_t(x)$ and 2 anchor samples $y_1', y_2' \sim \pi_{\text{anchor}}^t(x)$
  */ **Forward KL**                                                                          */
  Extract Best-of-2 sample: $y_{\text{Bo2}}' = \arg\max_{y' \in \{y_1', y_2'\}} r(y')$.
  Compute forward KL gradient:    $G_{\text{FW}}(x, \pi_t) = \nabla_{\pi_t} \log \pi_t(x, y_{\text{Bo2}}')$
  */ **Backward KL**                                                                          */
  Compute $r_{\text{J-BOND}}(x, y)$ according to Equation (17).
  Compute return: $R(x, y) = r_{\text{J-BOND}}(x, y) - (\log \pi_t(x, y) - \log \pi_{\text{anchor}}^t(x, y))$.
  [Optional] Compute baseline $B$, e.g. average return of the other generations in the batch.
  Compute backward KL gradient:    $G_{\text{BW}}(x, \pi_t) = \nabla_{\pi_t} \log \pi_t(x, y) \cdot (R(x, y) - B)$.
  */ **Additional KL regularization**                                                         */
  KL regularization gradient: $G_{\text{Reg}}(x, \pi_t) = \nabla_{\pi_t} \log \pi_t(x, y) \cdot (\log \pi_t(x, y) - \log \pi_{\text{anchor}}^t(x, y))$.
  */ **Overall policy update:  Jeffreys divergence + KL regularization**
*/
  Update policy weights $\theta_{t+1}$ with the overall stochastic gradient:

  $$-1 \cdot \mathbb{E}_{x \sim \mathcal{D}_t}[(1 - \beta) \cdot G_{\text{FW}}(x, \pi_t) + \beta \cdot G_{\text{BW}}(x, \pi_t) + \gamma \cdot G_{\text{Reg}}(x, \pi_t)]$$

  */ **Update moving anchor**                                                                 */
  Update anchor weights with EMA:    $\theta_{\text{anchor}}^{t+1} \leftarrow (1 - \eta) \cdot \theta_{\text{anchor}}^t + \eta \cdot \theta_{t+1}$

---

That is, generation $y$ receives a negative reward of $-\log(16)$ if it has worse reward than *both* anchor samples, while receives $0$ reward otherwise. The above definition is motivated by the following two main reasons:

(i) We negatively reward $y$ only if it is worse than both the anchor samples, to mimic the concavity of the ideal (and unknown) reward function $r_{\text{BOND}} = \log p_\leq(\cdot)$.

(ii) We choose value $-\log(16)$ to ensure that: $\mathbb{E}_{y_1', y_2' \sim \pi_{\text{anchor}}^t}[r_{\text{J-BOND}}(y)] = \log p_\leq(y)$ when $p_\leq(y) = 0.5$. The interested reader is referred to Appendix A.4 for a derivation and illustration of this fact. In words, the value $-\log(16)$ calibrates the reward function $r_{\text{J-BOND}}(\cdot)$ so that, in expectation under the 2 anchor samples, it matches with the ideal reward $\log p_\leq(\cdot)$ for generations $y$ that have median reward (i.e., when $p_\leq(y) = 0.5$).

**Exponential Moving Average (EMA) anchor.** An important component of J-BOND, which refines the vanilla iterative BOND of Section 4.3, is the use of an *Exponential Moving Average (EMA)* anchor. That is, instead of using a periodically updated anchor, we update the anchor weights $\theta_{\text{anchor}}^t$ *at each fine-tuning step* as a moving average of the policy weights $\theta_t$:

$$\theta_{\text{anchor}}^{t+1} \leftarrow (1 - \eta) \cdot \theta_{\text{anchor}}^t + \eta \cdot \theta_{t+1}. \tag{18}$$

Consistently with WARP (Ramé et al., 2024), we observed that this weight averaging procedure has a positive effect on training stability by reducing variance, and can improve the overall reward/KL trade-off of J-BOND. We provide an ablation in Section 6.

**Additional KL regularization.** Finally, we further regularize the policy to stay closer to the moving anchor via an extra[2] KL regularization term, modulated by a tunable hyperparameter $\gamma \geq 0$. The scope is to further stabilize the policy updates, viewing the overall operator as a constrained optimization one:

$$\pi_{t+1} = \arg\min_{\pi \in \Pi} J_{\text{effreys}}^\beta(\pi \| \text{Best-of-2}(\pi_{\text{anchor}}^t)) \quad + \quad \gamma \cdot \text{KL}(\pi_t \| \pi_{\text{anchor}}^t). \tag{19}$$

---

[2]Note that KL regularization is already present in the backward KL component of the Jeffreys divergence.

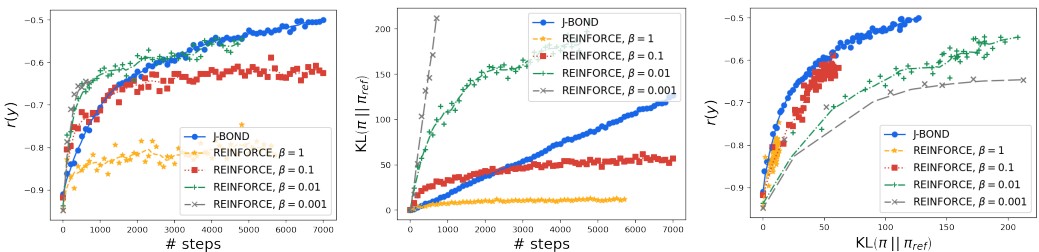

Figure 4: J-BOND ($\eta = 0.02$) for Gemma 7B, compared to standard REINFORCE (with KL-regularized objective of Equation (1)) with regularization strength $\beta_{\mathrm{RL}} \in \{0.001, 0.01, 0.1, 1\}$. J-BOND does not require committing to a specific regularization strength, but it continuously improves the reward displaying a stable and linear KL increase and a better reward/KL trade-off.

# 6  EXPERIMENTS

We test J-BOND on relevant use cases with the following main goals. First, we ablate and showcase important aspects of J-BOND: the benefits of the EMA anchor, and the effects of the anchor speed and the additional KL regularization. Then, we compare J-BOND to classical RLHF baselines using REINFORCE, demonstrating its efficacy and better performance.

**Setup.** We consider Gemma (2B and 7B) models (Gemma Team, 2024) which we aim to fine-tune into better conversational agents. For this task, we consider a set of conversational prompts $\mathcal{D}$, a reference policy $\pi_{\mathrm{ref}}$ previously supervised fine-tuned on similar prompts, and a large previously trained reward model $r(\cdot)$. We use a batch size of 128 and the Adam optimizer (Kingma & Ba, 2015) with learning rate $3e{-}6$ and 100 warm-up steps. For the Jeffreys divergence objective, we set $\beta = 0.5$ (we ablate different Jeffreys divercences in Appendix B.3).

**EMA vs. hard anchor updates.** We ablate the benefits of using an EMA moving anchor (Equation (18)) compared to the periodically updated anchor used in Section 4.3. For this, we run J-BOND with $\gamma = 0$ and EMA coefficient $\eta = 0.02$ on Gemma 7B, and compare it with its variant where the anchor is only updated every 50 steps. In Figure 10 (in Appendix B.4 due to space constraints), we report the average reward of the policy during training (left plot), the KL from the reference policy $\pi_{\mathrm{ref}}$ (middle plot), and the resulting reward/KL trade-off for the two runs. Both runs produce the same reward increase profile (this is not surprising since an EMA with $\eta = 0.02$ roughly corresponds to an update period of 50 steps) but, crucially, J-BOND *with an EMA anchor displays a significantly lower KL increase* and, as a result, a better reward/KL trade-off.

**Anchor speed and KL regularization.** We illustrate the effect of the anchor mixing parameter $\eta \in [0, 1]$ and the benefits of the additional KL regularization parameter $\gamma \geq 0$ introduced in Equation (19). For this, we consider Gemma 2B and first run J-BOND with $\gamma = 0$ and $\eta \in \{0.01, 0.05, 0.1\}$. In the left plot of Figure 11 (relegated to Appendix B.4) we report the average reward of the policy along training. This illustrates that *the larger the mixing parameter $\eta$ (i.e., the faster the anchor moves), the faster the reward increases*, as one could intuitively expect. Second, we fix $\eta = 0.05$ and run J-BOND with different regularization strengths $\gamma \in \{0, 0.5, 1, 2\}$. We plot the results in the middle and rightmost plots of Figure 11. As expected, *the larger the regularization $\gamma$, the more constrained are the policy updates and thus, *the slower the policy moves away from $\pi_{ref}$* (middle plot). Importantly, the right plot shows that such a regularization has a positive effect since it can ultimately improve the reward/KL trade-off.

**Comparison with standard RLHF.** We compare J-BOND against standard RLHF algorithms that aim at maximizing the KL-regularized objective of Equation (1). To optimize Equation (1), we use REINFORCE (Williams, 1992) with 2 policy samples per-prompt and a leave-one-out baseline (Ahmadian et al., 2024) for policy gradient advantages. For J-BOND we set the anchor mixing coefficient to $\eta = 0.02$. For REINFORCE, we test possible regularization strengths $\beta_{\mathrm{RL}} \in \{0.001, 0.01, 0.1, 1\}$. In Figure 4 we plot the average reward of the training policy (left plot) and its KL divergence from the reference $\pi_{\mathrm{ref}}$ (middle plot). As presumed, REINFORCE is quite sensitive to the regularization coefficient $\beta_{\mathrm{RL}}$: the larger the regularization, the lower the reward achieved by REINFORCE (and the lower the KL from $\pi_{\mathrm{ref}}$). This highlights a key advantage of J-BOND: it does not require committing

to a specific regularization level, but it continuously improves the reward displaying a stable and linear KL increase. Moreover, in the rightmost plot we plot the corresponding reward/KL trade-off showing that J-BOND produces a better reward/KL than all of the REINFORCE baselines. This translates into a higher quality (measured via side-by-side comparisons) and improved scores on popular real-world benchmarks, as we report in Appendix B.5.

## 7 RELATED WORK

**Best-of-N** was introduced in Stiennon et al. (2020) as a straightforward but costly inference method to optimize language generation against a given reward function. Further works established and refined an analytical form for the KL divergence against the reference (i.e., Bo1) policy (Hilton, 2023; Beirami et al., 2024), provided an estimator for the average Best-of-N reward (Nakano et al., 2021), made theoretical connections with KL-constrained RL (Yang et al., 2024) and provided scaling laws for Best-of-N alignment (Gao et al., 2023).

**Matching Best-of-N for improved alignment** is a strategy that was studied in different flavors in the literature. Dong et al. (2023) and Touvron et al. (2023) propose to fine-tune LLMs in a supervised fashion on Best-of-N data actually applying forward KL minimization. Concurrently to ours, Gui et al. (2024) proposes to mimic the Best-of-N policy by applying a combination of supervised fine-tuning on best responses and direct preference optimization on best-and-worst response pairs. The latter is similar to a common strategy in online preference optimization methods: Guo et al. (2024) use pairwise AI feedback on online generations to obtain online preferences that are then optimized, while Calandriello et al. (2024) use a dedicated preference reward model instead. Concurrently and closest to our work, Amini et al. (2024) also apply distribution matching in order to get the benefits of Best-of-N sampling with amortized cost. While their formalization is identical, we opt for a different divergence (i.e., Jeffreys) than the one they use (i.e., only backward KL), and propose an iterative procedure with dynamic anchor, which we show critical for optimal results. We compare J-BOND with the vBoN approach of (Amini et al., 2024) in Appendix A.5. Best-of-N can also be used for self-improvement in reward modeling, as evidenced in Pace et al. (2024).

**Using a contrastive advantage** is an option of J-BOND studied in prior works as well, which replaced a value estimate by the average Monte Carlo return of other samples. This was applied in the context of REINFORCE (Kool et al., 2019; Pinto et al., 2023), for online RLHF (Ahmadian et al., 2024), offline RLHF (Flet-Berliac et al., 2024) and preference optimization (Wu et al., 2024).

**Exponential moving average (EMA) of policy as reference** in regularization, which we use in J-BOND, is an increasingly popular option. While most alignment approaches use a static anchor, dynamic anchors bring the benefit of improving the flexibility of the policy space being explored (Munos et al., 2023; Gorbatovski et al., 2024; Ramé et al., 2024), with the caveat that too slow updates limit optimization and too fast updates hinder stability.

**Scaling post-training and iterated amplification**. BOND hinges on the idea of investing more resources during training to ensure that computational demands during inference remain low, a factor often overlooked in traditional scaling laws (Hoffmann et al., 2022). Specifically, BOND incorporates the principles of iterated amplification (Christiano et al., 2018; Cotra, 2018), where amplification in this context consists of producing multiple generations, comparing their rewards, and using these to iteratively improve the policy performance. In this regard, BOND is complementary to WARM (Ramé et al., 2024) and WARP (Ramé et al., 2024), which previously scaled post-training by training multiple reward models and policies, respectively.

## 8 CONCLUSION

We introduce BOND, a novel RLHF method that fine-tunes the policy via online distillation of the Best-of-N sampling distribution. We propose a concrete algorithm, J-BOND, that integrates multiple components to enhance its practicality and efficiency; Monte-Carlo quantile estimation, a combination between forward and backward KL divergence objectives, and an iterative procedure with an exponential moving average anchor. J-BOND improves the KL-reward Pareto front of solutions, and compares favorably against state-of-the-art baselines. We hope this work can help improve alignment of AI systems, making them safer and more reliable.

ACKNOWLEDGEMENTS

We thank Daniele Calandriello for insightful comments, as well as Gil Shamir, Bilal Piot, and Remi Munos for helpful discussions.

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

## A SUPPORTING RESULTS AND DERIVATIONS

### A.1 PROOF OF THEOREM 1

Consider $N$ random generations $y_1, y_2, \ldots, y_N$ from $\pi_{\text{ref}}$ and an arbitrary generation $y$ among them. Let $A_i(y)$ denote the event that $y$ is the best sample (i.e., $r(y) \geq r(y_i)$ for all $i$) and that $i$ is the lowest index for which $y_i = y$. It is trivial to see that the the events $\{A_i(y)\}_{i=1,2,\ldots,N}$ are disjoint and that their union corresponds to $y$ being selected by Best-of-N sampling.

The event $A_i(y)$ occurs if and only if three conditions are met: $r(y_j) < r(y)$ for all $j < i$, $y_i = y$, and $r(y_j) \leq r(y)$ for all $j < i$. This allows us to derive the likelihood of the event $A_i(y)$:

$$
\mathbb{P}[A_i(y)] = \left( \prod_{j=1}^{i-1} \mathbb{P}[r(y_j) < r(y)] \right) \times \pi_{\text{ref}}(y) \times \left( \prod_{j=i+1}^{N} \mathbb{P}[r(y_j) \leq r(y)] \right)
$$
$$
= p_<(y)^{i-1} \times \pi_{\text{ref}}(y) \times p_\leq(y)^{N-i-1}.
$$

The likelihood that Best-of-N sampling selects the generation $y$ is then given by

$$
\pi_{\text{BoN}}(y) = \sum_{i=1}^{N} \mathbb{P}[A_i(y)]
$$
$$
= \sum_{i=1}^{N} \left[ p_<(y)^{i-1} \times \pi_{\text{ref}}(y) \times p_\leq(y)^{N-i} \right]
$$
$$
= \pi_{\text{ref}}(y) \times \sum_{i=1}^{N} \left[ p_<(y)^{i-1} \times p_\leq(y)^{N-i} \right]
$$
$$
= \pi_{\text{ref}}(y) \times p_\leq(y)^{N-1} \times \sum_{i=1}^{N} \left[ \frac{p_<(y)}{p_\leq(y)} \right]^{i-1}. \quad \square
$$

### A.2 LINK TO THE CONTINUOUS CASE

A noteworthy observation is that we can relate the Best-of-N expression to the case of a continuous distribution, in which case the term (B) in Equation (4) is constant and equal to $N$ (which is intuitively natural as $p_<(y)$ and $p_\leq(y)$ have the same value in this case).

Indeed, recall that the probability for a sequence $y$ to be drawn from the Best-of-N distribution is

$$
\pi_{\text{BoN}}(y) = \pi_{\text{ref}}(y) \times \underbrace{p_\leq(y)^{N-1}}_{\text{(A)}} \times \underbrace{\sum_{i=1}^{N} \left[ \frac{p_<(y)}{p_\leq(y)} \right]^{i-1}}_{\text{(B)}}. \tag{20}
$$

Here, $y$ is a discrete variable, as it lives in $[1; T]^L$ where $T$ is the number of tokens and $L$ is the maximum length of a sequence.

Now, we show why Equation (20) matches the classic formula for the max of $N$ continuous variables. Formally, let $X$ be a real valued random variable with density $f_X$ and a cumulative distribution function $F_X$. Taking $Y_1, \ldots Y_N$ i.i.d. variables with the same density, define $X_N = \max\{Y_1, \ldots Y_N\}$ as the maximum over the $N$ variables. Then, we have that

$$
F_{X_N}(y) = \mathbb{P}(Y_1 \leq y, \ldots Y_N \leq y) = F_X(y)^N, \tag{21}
$$

and thus

$$
f_{X_N}(y) = f_X(y) F_X(y)^{N-1} N. \tag{22}
$$

In Equation (22), we recognize the Best-of-N formula in the case where the correction factor (B) is $N$. For the term (A), $F_X(y)$ plays the role of $p_\leq(y)$, as by definition $F_X(y) = \mathbb{P}(X \leq y)$. Finally, $f_X(y)$ is the density of $X$, which is analogous to the probability $\pi_{\text{ref}}(y)$ in the discrete case.

### A.3 BACKWARD KL AND POLICY GRADIENT EQUIVALENCE

We formally show the analogy between the gradient of the backward KL divergence of Equation (14) and the standard (e.g.,REINFORCE (Williams, 1992)) policy gradient of a KL-regularized RLHF problem with equivalent reward $r_{\text{BOND}}$ and regularization $\beta_{\text{BOND}}$.

The exact backward KL gradient can be derived as:

$$
\begin{aligned}
\nabla_\pi \text{KL}(\pi \,\|\, \pi_{\text{BoN}}) &= \nabla_\pi \mathbb{E}_{y\sim\pi}[\log \pi(y) - \log \pi_{\text{BoN}}(y)] \\
&= \nabla_\pi \sum_y \pi(y)(\log \pi(y) - \log \pi_{\text{BoN}}(y)) \\
&= \sum_y \nabla_\pi \pi(y)(\log \pi(y) - \log \pi_{\text{BoN}}(y)) + \pi(y)\nabla_\pi \log \pi(y) \\
&= \sum_y \pi(y)\nabla_\pi \log \pi(y)(\log \pi(y) - \log \pi_{\text{BoN}}(y)) + \pi(y)\nabla_\pi \log \pi(y) \\
&= \mathbb{E}_{y\sim\pi}[\nabla_\pi \log \pi(y)(\log \pi(y) - \log \pi_{\text{BoN}}(y)) + \nabla_\pi \log \pi(y)] \\
&= \mathbb{E}_{y\sim\pi}[\nabla_\pi \log \pi(y)(\log \pi(y) - \log \pi_{\text{BoN}}(y))].
\end{aligned}
$$

Above, we have used the product rule of gradient, the rule $\nabla_\pi \pi(y) = \pi(y)\nabla_\pi \log \pi(y)$ and the fact that $\mathbb{E}_{y\sim\pi}\nabla_\pi \log \pi(y) = 0$.

*Equivalence with Policy Gradient RL.* As anticipated, one can verify that descending the above gradient is equivalent – up to a constant scaling – to running the RL policy gradient REINFORCE algorithm on the RL objective of Equation 1 with $r = r_{\text{BOND}}$ and $\beta_{\text{RL}} = \beta_{\text{BOND}}$. Indeed, we can use the expression for $\pi_{\text{BoN}}$ to break down the above gradient into:

$$
\begin{aligned}
&\mathbb{E}_{y\sim\pi}[\nabla_\pi \log \pi(y)(\log \pi(y) - \log \pi_{\text{BoN}}(y))] \\
&= \mathbb{E}_{y\sim\pi}\left[\nabla_\pi \log \pi(y)\left(\log \pi(y) - \log \pi_{\text{ref}}(y) - (N-1)\log p_\leq(y) - \log \sum_{i=1}^N \left[\frac{p_<(y)}{p_\leq(y)}\right]^{i-1}\right)\right] \\
&= \mathbb{E}_{y\sim\pi}\left[\nabla_\pi \log \pi(y)\left(\log \pi(y) - \log \pi_{\text{ref}}(y) - \frac{r_{\text{BOND}}(y)}{N-1}\right)\right] \\
&= -(N-1)\underbrace{\mathbb{E}_{y\sim\pi}[\nabla_\pi \log \pi(y)(r_{\text{BOND}}(y) - \beta_{\text{BOND}}(\log \pi(y) - \log \pi_{\text{ref}}(y)))]}_{\text{gradient used by REINFORCE}}.
\end{aligned}
$$

### A.4 DERIVATION OF J-BOND REWARD

Here we provide a theoretical explanation behind the design of the J-BOND reward function discussed in Section 5:

$$
r_{\text{J-BOND}}(y) = \begin{cases} -\log(16) & \text{if} \quad r(y) \leq \min\{r(y_1'), r(y_2')\} \\ 0 & \text{otherwise} \end{cases}.
$$

Recall that $r_{\text{J-BOND}}(\cdot)$ is meant to approximate the true reward function $r_{\text{BOND}}(y) = \log p_\leq(\cdot)$ which is unknown since $p_\leq(\cdot)$ in general requires knowing the reward distribution of $\pi_{\text{anchor}}^t$ (in J-BOND, we only take 2 samples $y_1', y_2' \sim \pi_{\text{anchor}}^t$).

As mentioned in Section 5, we designed $r_{\text{J-BOND}}(\cdot)$ to assign a negative reward only if sample $y$ is worse than both the anchor samples, to mimic the concavity of the log quantile $\log p_\leq(\cdot)$. In practice, we did not observe gains when rewarding also the intermediate case. The particular choice of value $-\log(16)$ is motivated by the following main reason.

We want that, when sample $y$ has median reward compared to the anchor rewards' distribution (i.e., $p_\leq(y) = 0.5$), then – in expectation – $r_{\text{J-BOND}}(y)$ coincides with the true reward $r_{\text{BOND}}(y) = \log p_\leq(\cdot) = \log(0.5)$. For this purpose, let us consider the parametrized function:

$$
r_{\text{J-BOND}}^\alpha(y) = \begin{cases} \alpha & \text{if} \quad r(y) < \min\{r(y_1'), r(y_2')\} \\ 0 & \text{otherwise} \end{cases}.
$$

Note that the stochasticity of $r^\alpha_{\text{J-BOND}}(y)$ is due to the 2 random anchor samples $y'_1, y'_2$ and its expectation can be computed as:

$$\mathbb{E}_{y'_1, y'_2 \sim \pi^t_{\text{anchor}}}[r^\alpha_{\text{J-BOND}}(y)] = \alpha \cdot \mathbb{P}[\{r(y'_1) > r(y)\} \cap \{r(y'_2) > r(y)\}] + 0 \cdot \mathbb{P}["\text{otherwise}"] \quad (23)$$

$$= \alpha \cdot (1 - p_\le(y))^2, \quad (24)$$

where we have used the definition of $p_\le(y) = \mathbb{P}_{y' \sim \pi^t_{\text{anchor}}}[r(y') \le r(y)]$. Using the expression above, we can find the $\alpha$ for which $\mathbb{E}_{y'_1, y'_2 \sim \pi^t_{\text{anchor}}}[r^\alpha_{\text{J-BOND}}(y)] = r_{\text{BOND}}(y)$ when $p_\le(y) = 0.5$:

$$\alpha \cdot (1 - 0.5)^2 = \log(0.5) \quad \rightarrow \alpha = -\log(16).$$

We illustrate this in Figure 5, where we plot the expected $r_{\text{J-BOND}}(y)$ reward and the true reward $r_{\text{BOND}}(y)$ as a function of $p_\le(\cdot)$.

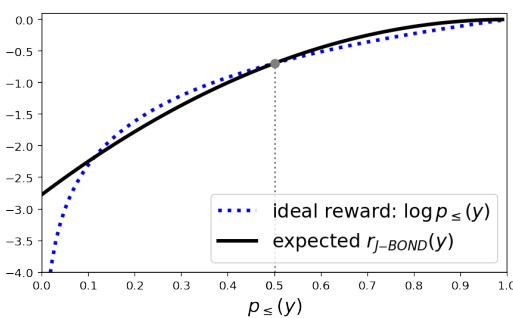

Figure 5: Expected value of the J-BOND reward, i.e., $\mathbb{E}_{y'_1, y'_2 \sim \pi^t_{\text{anchor}}}[r_{\text{J-BOND}}(y)]$ (see Equation (23)), compared to the true (unknown) reward $\log p_\le(y)$, as a function of the quantile $p_\le(y)$. By design of $r_{\text{J-BOND}}$, the two curves coincide when $p_\le(y) = 0.5$, i.e., when $y$ has median reward w.r.t. the anchor distribution.

### A.5 COMPARISON WITH VBON (AMINI ET AL., 2024)

In this section we compare J-BOND with the concurrent VBON approach of Amini et al. (2024). Amini et al. (2024) consider the same BOND objective of Section 3.2 with the main difference that only backward KL divergence is considered. Thus, according to the equivalence drawn in Equation (15), their approach can be seen as using standard RLHF algorithms to optimize the equivalent reward $r_{\text{BOND}}$ of Equation (8) with regularization strength $\beta_{\text{BOND}} = \frac{1}{N-1}$. Below, we summarize the main differences with respect to J-BOND:

(i) VBON considers only backward KL divergence, while J-BOND uses Jeffreys divergence.

(ii) VBON requires setting a fixed $N$ in advance (similar to regularization in standard RLHF), while J-BOND continuously improves the reward thanks to its iterative nature.

(iii) To compute $r_{\text{BOND}}(y)$, the vanilla version of VBON presented in (Amini et al., 2024) requires estimating the quantile $p_\le(y)$ with several MC samples. Unfortunately, this is only doable when the number of prompts is small and the estimation can be done offline, but it is prohibitive in more complex experimental setups with millions of prompts.

To overcome challenge (iii), we consider a variation of VBON where $r_{\text{BOND}}(y)$ are estimated via the exact same crude estimate used in J-BOND, based on 2 anchor samples (Equation (17)). We consider the Gemma 7B finetuning experiment of Section 6 and compare J-BOND to VBON for $N \in [4, 8, 64, 128, 512]$. The corresponding results are reported in Figure 6. Unlike VBON where the reached reward and KL depend on the value of $N$ defined in advance, J-BOND displays a stable KL and reward increase yielding a better reward/KL trade-off. This is attributed to its iterative approach and to the fact that both backward and forward KL divergences are minimized.

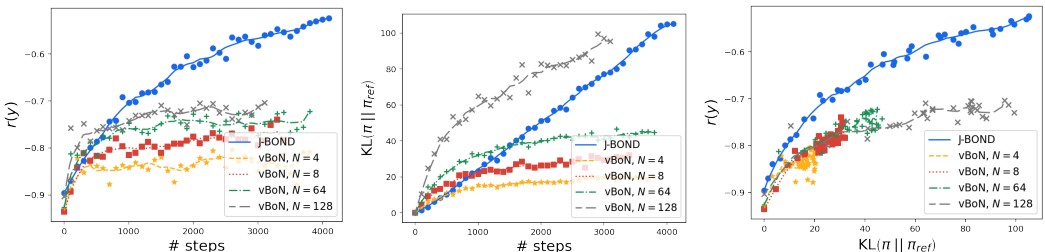

Figure 6: Comparison of J-BOND and our implementation of vBON (Amini et al., 2024) for finetuning Gemma 7B. Compared to (Amini et al., 2024), we approximate the backward KL objective via the crude reward estimate of Equation (17) and optimize it with standard policy gradient. This is because estimating the actual BOND reward of Equation (8) requires several anchor samples and is not feasible in our online setup. Unlike vBON, J-BOND does not require setting a fixed $N$ in advance and produces a stable and improved reward/KL trade-off.

# B    ADDITIONAL PLOTS AND EXPERIMENTS

## B.1    BOND WITH JEFFREYS DIVERGENCE OBJECTIVE

In Figure 7, we report the results concerning the experimental setup described in Section 3.2, i.e. we run BOND with $J^{\beta}_{\text{effreys}}$ objective and $\beta \in \{0, 0.5, 1\}$. The Jeffreys divergence ($\beta = 0.5$) allows to minimize both divergences from $\pi_{\text{BoN}}$ (left and middle plot), compared to when solely the backward ($\beta = 1$) or the forward ($\beta = 0$) KL divergence is minimized.

## B.2    LEARNED QUANTILE MODELS

Monte-Carlo quantile estimation (Section 4.1) approximates the reward quantiles by sampling multiple times from the reference policy $\pi_{\text{ref}}$, for each observed context. While we found it to be very simple and effective, it may require many sample for an accurate quantile estimation and, in addition, it does not exploit any information about the given context. For instance, assuming we have a good quantile estimation for context $x$ and are presented a new context $x'$. MC quantile estimates treat $x'$ independently from $x$, although they may have very similar reward quantiles.

Motivated by this, in this section we explore an alternative approach that aims at *learning* a context-dependent quantile estimator $\widehat{p_{\leq}}_{\theta}(\cdot)$, parametrized by parameter $\theta$. The idea is to view quantile $p_{\leq}(y)$ as the parameter of a Binomial random variable $Z$ where $Z = \mathbb{I}\{r(y_i) \leq r(y)\}$ for $y_i \sim \pi_{\text{ref}}$. Under such a view, we can interpret $\widehat{p_{\leq}}_{\theta}(\cdot)$ as the output of a binary classifier and train it via maximum likelihood estimation using the standard binary cross-entropy loss (Cover, 1999):

$$L(\theta, x, y) = -\mathbb{E}_{y' \sim \pi_{\text{ref}}(x)}\left[\log \widehat{p_{\leq}}_{\theta}(x, y)\mathbb{I}_{\{r(x,y') \leq r(x,y)\}} + \log(1 - \widehat{p_{\leq}}_{\theta}(x, y))\mathbb{I}_{\{r(x,y') > r(x,y)\}}\right].$$
$$(25)$$

We test such an approach in the abstractive summarization task considered in Section 4. We parametrize $\widehat{p_{\leq}}_{\theta}(\cdot)$ with a LLM initialized as $\pi_{\text{ref}}$ and fine-tuned using the loss of Equation (25). Simultaneously, the policy $\pi_t$ is fine-tuned using BOND and utilizing $\widehat{p_{\leq}}_{\theta_t}(\cdot)$ as quantile estimator at each step. Notably, we approximate the expectation in Equation (25) via a *single* sample from $\pi_{\text{ref}}$ for each prompt in the batch. In Figure 8 we report the backward and forward KL divergences between the training policy and the $\pi_{\text{BoN}}$ distribution as well as the average log quantiles, and compare them to the ones obtained when running BOND with MC quantile estimation. Note that in both cases, $\pi_{\text{BoN}}$ is approximated by 32 MC samples during evaluation. When using the learned quantile model $\widehat{p_{\leq}}_{\theta}(\cdot)$, we observe BOND achieves very comparable KL divergences and log quantiles compared to using MC quantile estimaton. This illustrates that the use of learned quantiles is valid and promising, potentially offering interesting computational advantages in situations where, e.g., $\widehat{p_{\leq}}_{\theta}(\cdot)$ can be re-used or learned offline with a fixed sample budget.

Finally, we remark that the explored approach is quite naive, and alternative learned quantile models can definitely be derived, e.g., further enforcing ordering in the predicted quantiles, using quantile regression (Dabney et al., 2017), or assuming a pre-specified (e.g., Gaussian) rewards' distributions.

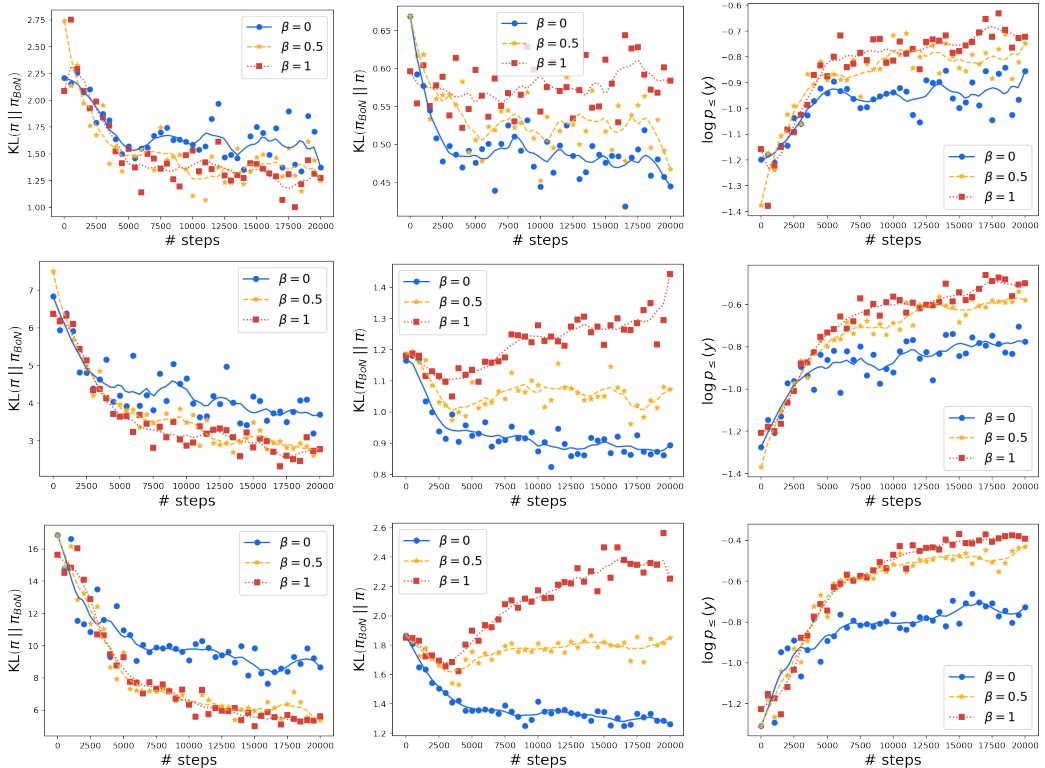

Figure 7: BOND with $N = 4$ (top row), $N = 8$ (middle row), and $N = 16$ (bottom row), and different values of $\beta$ for the Jeffreys divergence objective (*cf.* Equation (11)). When using $\beta = 0.5$ (Jeffreys divergence), BOND minimizes both backward (left plots) and forward (middle plots) KL divergences from $\pi_{\text{BoN}}$, achieving best of both objectives ($\beta = 0$ and $\beta = 1$). Moreover, it optimizes the reward quantiles (right plots) significantly more than when using $\beta = 0$.

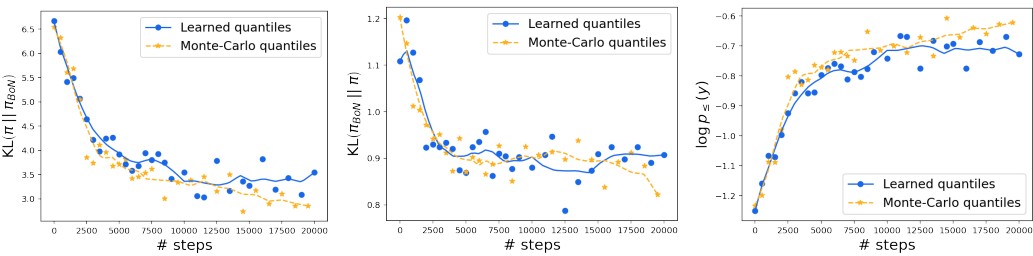

Figure 8: BOND (with $N = 8$) using MC quantile estimates vs. a Learned quantile model.

### B.3 J-BOND ABLATIONS: JEFFREYS DIVERGENCE

Here we ablate the effect of parameter $\beta$ in J-BOND, i.e., we test J-BOND with different Jeffreys divergences. We consider the Gemma 7B finetuning experiment of Section 6 and run J-BOND with different choices for $\beta \in [0, 0.25, 0.5, 0.75, 1]$. We report the obtained results in Figure 9. J-BOND with $\beta = 0.5$ (in red) displays the best reward/KL trade-off, highlighting the importance of minimizing Jeffreys divergences as opposed to only backward ($\beta = 1$) or forward ($\beta = 0$) KL. The ablation complements the one of Appendix B.1 showing that Jeffreys divergence as objective is also beneficial for J-BOND to achieve best Pareto solutions.

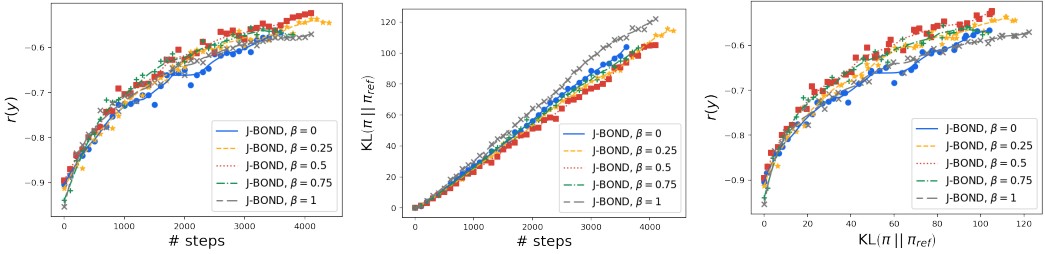

Figure 9: J-BOND with different Jeffreys divergences (parametrized by $\beta$) when finetuning Gemma 7B. Employing Jeffreys divergence as an objective, i.e., $\beta \in (0,1)$ leads to a better reward/KL trade-off (rigthmost plot) than when only backward ($\beta = 1$) or forward ($\beta = 0$) KL are considered. In this example, the best reward/KL trade-off is achieved by $\beta = 0.5$.

### B.4   J-BOND ABLATIONS: EMA ANCHOR AND KL REGULARIZATION

Here we provide the plots associated to the J-BOND ablations discussed in Section 6. In particular, Figure 10 illustrates the benefit of the EMA anchor compared to periodic anchor updates. Moreover, Figure 11 illustrates the role of the EMA mixing parameter $\eta$ and the regularization parameter $\gamma$ in J-BOND. See Section 6 for the detailed setups and related discussions.

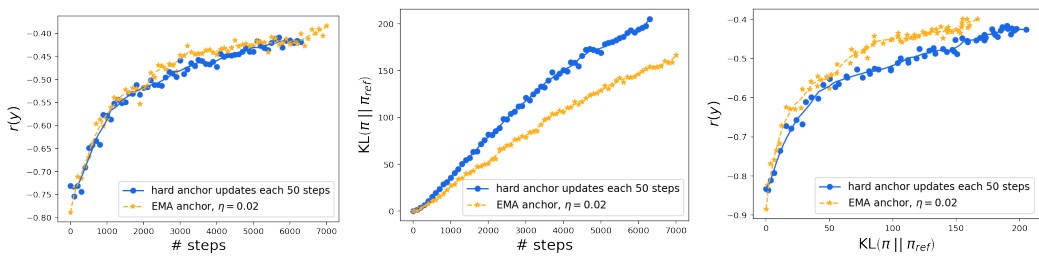

Figure 10: J-BOND with periodic anchor updates (every 50 steps) vs. EMA anchor ($\eta = 0.02$) on Gemma 7B. While attaining the same reward (left), using the EMA anchor displays a significantly lower KL than the reference policy (middle) and thus a better reward/KL trade-off (right).

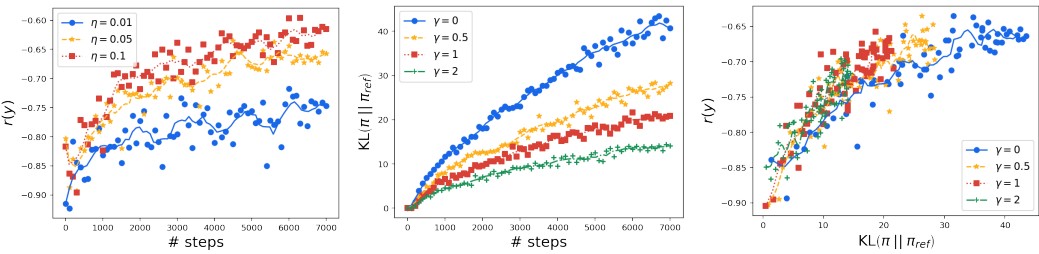

Figure 11: J-BOND: the role of the EMA mixing parameter $\eta$ and regularization $\gamma$. The left plot shows J-BOND with $\gamma = 0$, $\eta \in \{0.01, 0.05, 0.1\}$: the larger the $\eta$ the faster the average reward increases. The middle and right plots show J-BOND with $\eta = 0.05$ and $\gamma \in \{0, 0.5, 1, 2\}$: the larger the regularization $\gamma$, the slower the policy moves away from $\pi_{\text{ref}}$, improving the reward/KL trade-off.

### B.5   DOWNSTREAM EVALUATIONS

We report downstream evaluations of the Gemma 7B policies finetuned with J-BOND and the REINFORCE baselines considered in Section 6. In particular, we report:

- Side-by-side comparisons: For each trained policy we generate candidate answers on a held-out collection of prompts. We compare such answers with the ones generated

by Mistral (Jiang et al., 2023) and Mixtral (Jiang et al., 2024) models. We use Gemini (Gemini Team, 2023) as a judge and for each comparison we assign a score in $\{-1, 5, -1, -0.5, 0, 0.5, 1, 1.5\}$ ranging from "much worse" $(-1.5)$ to "much better" $(1.5)$. We then report the average score over all prompts.

- Zero-shot performance on popular benchmarks including: GPQA (Rein et al., 2024), GSM8K (Cobbe et al., 2021), MATH (Hendrycks et al., 2021) and Big Bench Hard (BBH) (Suzgun et al., 2022) to test our policies on different capabilities.

We evaluate multiple checkpoints for J-BOND and REINFORCE and in Table 1 report the numbers corresponding to the best ones. J-BOND consistently and significantly outperforms the REINFORCE (standard RLHF) baseline both in terms of quality, as measured by the side-by-side comparisons, and in terms of accuracy on real-world benchmarks.

| Side-by-side comparisons | | | |
| --- | --- | --- | --- |
| | Mistral 7B v1 | Mistral 7B v2 | Mixtral 8x7B |
| Standard RLHF | 0.34 | 0.16 | 0.04 |
| **J-BOND** | **0.36** | **0.17** | **0.07** |

| Benchmarks | | | |
| --- | --- | --- | --- |
| | GPQA | GSM8K | MATH | BBH |
| Standard RLHF | 26.6 | 45.4 | 26.8 | 53.3 |
| **J-BOND** | **31.2** | **54.2** | **28** | **54.1** |

Table 1: Downstream evaluations of Gemma 7B finetuned with J-BOND compared to standard RLHF, i.e. REINFORCE with leave-one-out baseline (Ahmadian et al., 2024).

