# OpenReview forum: "BOND: Aligning LLMs with Best-of-N Distillation"
_ICLR.cc/2025/Conference — ICLR 2025 Poster_

### Official Review · Reviewer_NVi9 · 2024-10-16

**Soundness:** 4
**Presentation:** 2
**Contribution:** 4
**Rating:** 8
**Confidence:** 4

**Summary:**

This article models the alignment of LLMs as a distribution-matching problem. Specifically, it considers the distribution induced by Best-of-N sampling as the target distribution and optimizes the Jeffreys divergence with respect to it for balancing the characteristics of forward & backward KL-based optimization. Additionally, this work derives an iterative form, updating the reference model during training.

**Strengths:**

1. This work implements the LLM alignment problem through Best-of-N distillation, which can be a sound direction for the development of algorithms in this field.
2. This work formulates and discusses the BoN distribution and its relationship with the general RLHF target.
3. This work proposes to utilize Jeffreys divergence to balance the mode-covering and mode-seeking behavior introduced by forward- and backward-KL- KL optimization.
4. This work further integrates their method with EMA techniques and proposes an iterative version.

**Weaknesses:**

1. My biggest concern is the absence of direct comparison with close-related works http://arxiv.org/abs/2407.06057, http://arxiv.org/abs/2406.00832. It would be much more convincing if there is a clear comparison like Figure 1 in http://arxiv.org/abs/2407.06057, rather than a simple baseline REINFORCE presented in Figure 7 in this work.
2. More discussion about why BoN distribution as the target can be included.
3. Other possible additional analyses are discussed in the Questions.

**Questions:**

#### Question 1:
Line 298, why is the proposed approach "equivalently to distilling a Best-of-$N^M$ of the initial distribution $\pi_{ref}$"? Is this a qualitative or quantitative statement?
#### Question 2:
Figure 4, seems like a linear relationship between # steps and r(y) for iterative BOND.
For Figure 5, it seems like another kind of trending between $r(y)$ and # steps or $KL(\pi||\pi_{ref})$.
Figures 6 and 7, present a log-like trend between the KL and reward similar to the REINFORCE algorithm.
A similar trend is also observed in http://arxiv.org/abs/2406.00832 and http://arxiv.org/abs/2407.06057.
As discussed in http://arxiv.org/abs/2204.05862 and my experience, there can be an approximately linear relationship between $\sqrt{D_{KL}}$  and $r$ for BoN sampling.
It would be interesting if the author could provide some empirical or theoretical intuition about such a relationship.
Does this indicate that even though performing a BoN distribution match, it is still more similar to the general policy-gradient RL algorithm (which may try to match another distribution)?
#### Question 3:
Why is there an approximately linear relationship between #steps and KL in, as presented in Figures 5 and 6 for the BOND algorithm?
For the REINFORCE algorithm, it seems a quite different trend.
#### Question 4:
Figure 4 presents consistent trending between KL and $\log_{p\le(y)}$ across varying $N$ and algorithm, which is quite interesting.
Is there any explanation for this phenomenon?
#### Question 5:
Is there any analysis or comparison of the reward overoptimization of this algorithm?

---

> ### Author Response · Authors · 2024-11-21
>
> Thank you for the positive feedback and the constructive comments. We address your concerns below.
>
> As mentioned in the general response, we have included additional ablations of J-BOND and direct comparisons w.r.t concurrent work of (Amini et al. 2024) in Appendix A.5.
> Quoting from the general response: In short, unlike J-BOND, vBoN requires fixing a regularization strength in advance (via the parameter $N$) and –moverover– considers only backward KL minimization which in (Amini et al. 2024) is estimated using several MC samples. Instead, J-BOND is designed to require only 2 anchor samples, optimizes Jeffreys divergence, and continuously optimize rewards achieving better reward/KL pareto solutions.
>
> Regarding comparisons with BoN, in Figure 7 (Appendix B.1) we compute actual KL distances between BOND policies and the BoN distribution. This can be done leveraging our analytical expression of Theorem 1 and using several MC samples to estimate quantiles. The goal is to show that our BOND objective (and its approximate gradients) are effective in terms of distilling the BoN distribution. When moving to Gemma fine tuning experiments, BoN sampling becomes intractable since it would require $N$ of order of thousands (if not more) to achieve the same rewards of J-BOND and the other RLHF baselines.
>
> To answer your questions:
>
> **Q1**: The statement is rather qualitative and abstract, in that it assumes perfect convergence of the BOND operator. However, assuming applying BOND to $\pi_{ref}$ *exactly* outputs the Best-of-$N$($\pi_{ref}$) distribution, then applying BOND again to such a distribution we are turning it into Best-of-$N^2$($\pi_{ref}$), and so on…
>
> **Q2,3&4**: We thank the reviewer for the questions. Indeed, it is quite interesting to analyze the rates of growth of KL, log quantiles, and rewards of J-BOND in comparison with the different baselines. First we would like to summarize two well-known results about BoN sampling:
> 1. The KL between BoN sampling and the reference policy is upper bounded by $log(N) - (N-1)/N$, i.e. it grows logarithmically with $N$. [Beirami et al. 2024]
> 2. The win rate (i.e. $p_\leq(y)$) between BoN sampling and the reference policy is $N/(N+1)$, *cf.* e.g. [Amini et al. 2024].
>
> The above two points are explainable of the trend curves associated to (iterative) J-BOND:
> For J-BOND, each step t can be seen as distilling the Best-of-$N^t$ distribution (see **Q1**). Thus, the KL vs. steps $t$, grows as $O(log(N^t)) = O(t)$, i.e. linearly, while the log quantiles grow sublinearly as $O(N^t/(N^t + 1))$. This is exactly visualized in Figures 3 and 4 and is an indication that, even with the several made approximations, J-BOND metrics are behaving according to theory. In terms of rewards, we believe there is no relationship that can be theoretically drawn, especially because J-BOND optimizes log quantiles and not the rewards directly. The difference between the reward trends of Figure 3 and 4 can simply be attributed to the different tasks (XSum vs. Gemma fine tuning) and reward models.
>
> [Beirami et al. 2024]: Theoretical guarantees on the best-of-n alignment policy
>
> **Q5**: We believe the Reward/KL tradeoff is quite a meaningful metric when it comes to reward overoptimization. Achieving better rewards without incurring a too high KL ensures the policy is not degenerating into exploitative behaviors of the reward model. This is what happens when, e.g. the tradeoff parameter $\beta$ of standard REINFORCE baselines is set too low. On the contrary, the fact that J-BOND displays a better and stable reward/KL pareto curve is a good indication of reward hacking being mitigated. Note that reward hacking is unavoidable after a certain KL due to the reward model becoming out-of-distribution. Finally, we note that J-BOND displaying such a better reward/KL tradeoff is quite interesting and encouraging, given that only log quantiles (i.e. generations’ rank) are optimized by the algorithm and not the actual rewards.
>
> We hope the above points clarify all of the reviewer’s concerns. We are happy to expand them further if needed.

---

> ### Comment · Reviewer_NVi9 · 2024-11-23
>
> Thank the authors for addressing my questions. I find the analysis proposed by the authors to be interesting and capable of explaining the observed training curves. Although the authors did not provide a more detailed explanation regarding the reward curve, I agree that this depends on the characteristics of both the RM and the initial policy. Despite not comparing all the baselines I listed, I believe the authors' analysis of (iterative) J-BOND and the alignment between theoretic analysis and experiment phenomenon convinces me of the algorithm's effectiveness and its novelty in comparison to other works on distilling BoN distributions. Therefore, I decide to raise my rating and score of the contribution.

---

### Official Review · Reviewer_rggM · 2024-10-27

**Soundness:** 3
**Presentation:** 2
**Contribution:** 3
**Rating:** 6
**Confidence:** 4

**Summary:**

The paper is essentially distilling inference-time best-of-N sampling into training time. Specifically, the authors propose to train the policy to match the best-of-N distribution (which is an analytical form derived by the authors). The distribution matching is done through minimizing a combination of forward and backward KL. The behavior of J-BOND appears better for reward vs. num steps as well as KL(policy || ref policy) vs. num steps, compared to REINFORCE with various beta values.

**Strengths:**

The motivation of distillation is great and this direction should be deeply explored. The paper is written carefully. The algorithm is clearly written. I checked the derivations and they seem correct so far.

REINFORCE is a natural baseline and the authors have attempted multiple beta values for the baseline.

**Weaknesses:**

Have the authors tried reward shaping techniques for RLHF baseline, e.g., making the reward values more peaky – either very large or very small?

I’d appreciate a more comprehensive discussion on how much the authors expect this technique to benefit downstream tasks.

It’ll be great if the authors can include more discussion on whether the baseline B is important or not in the algorithm.

What ablations on beta and gamma in Algorithms 2 (balancing among forward KL, backward KL, additional regularization) would likely benefit downstream tasks more? It's still unclear to me why we want to put an equal/large weight on backward KL. More motivation would be nice.

**Questions:**

Would the curves in Figure 4(a) converge to similar spots if you run the algorithms long enough?

Do the authors expect that the conclusions would be similar for much larger models?

Writing:
- pi(x,y) makes it look like a joint distribution; do the authors mean pi(y|x)?
- 3e-6 means 0.000003; 3e^(-6) means 0.007. Which one are you referring to?

---

> ### Author Response · Authors · 2024-11-21
>
> Thank you for reviewing our work and providing valuable feedback. Below, we respond to the main points and questions raised.
>
> Reward shaping can definitely improve the performance of standard RLHF algorithms, as demonstrated in several recent works. Though, we believe BOND provides a more principled alternative compared to possible reshaping approaches as it only relies on generations’ rank.
> Also, we note that the superior performance of BOND is not only attributed to optimizing a transformed reward (backward KL component), but also to forward KL minimization and by the iterative distillation procedure.
>
> In terms of impact on downstream tasks, we believe J-BOND provides two main benefits: 1) because it does not rely on reward values (but only on generations’ rank), the resulting policy should be less prone to reward optimization and thus of higher quality. 2) because of the iterative procedure, J-BOND fine tuning is more stable than standard baselines (e.g., Figure 4) and can return a whole spectrum of policies (at different KL divergences) that one could evaluate on the downstream task of interest. As mentioned in the general response, in Appendix B.5 we have reported downstream evaluations of the fine tuned policies showing that J-BOND achieves higher quality (measured via side-by-side comparisons) and significantly higher benchmark scores.
>
> We found reward baselines to be of high importance (as typical in RL and RLHF literature). Moreover, we have performed additional ablations (see general response) illustrating the effect of Jeffreys divergence in J-BOND. These are reported in Appendix B.3.
> The results complement the ablation performed in Appendix B.1 and highlight the fact that both forward and backward KL components are crucial for achieving best reward/KL tradeoffs.
> Intuitively, the forward KL component makes sure the policy covers all the modes of the BoN distribution, while the backward KL makes sure the policy is not over-dispersed and makes it peaky on certain modes.
>
> Finally, we provide answers to the raised questions:
>
> > Would the curves in Figure 4(a) converge to similar spots if you run the algorithms long enough?
>
> In Figure 4(a), the curves with a fixed N (i.e. $N=4,8,16$) are sublinear and thus will asymptote to different reward values, depending on the fixed $N$. Instead, the iterative BOND with $n=2$ is a slower version of the iterative BOND with $n=4$ so it will eventually reach the same reward values. That is, unlike BOND with a fixed $N$, iterative BOND improves the rewards continuously.
>
> > Do the authors expect that the conclusions would be similar for much larger models?
>
> Yes, we expect similar conclusions for larger models. Arguably, larger models could be more prone to reward over-optimization while BOND could mitigate this via the robust BoN distillation objective.
>
> > pi(x,y) makes it look like a joint distribution; do the authors mean pi(y|x)?
>
> Yes, we mean the conditional distribution. We’ll improve the notation accordingly.
>
>
> > 3e-6 means 0.000003; 3e^(-6) means 0.007. Which one are you referring to?
>
> Thanks for spotting the typo, indeed we mean 3e-6.
>
> We hope the above points clarify all of the Reviewer’s concerns. We are happy to expand them further if needed.

---

> > ### Comment · Reviewer_rggM · 2024-11-26
> >
> > Thank you for the response and the additional ablations (e.g., Appendix B.3). I'm increasing my score to 6 but it'll be great if the paper can include a more detailed discussion on the relationship between reward shaping techniques and BOND (and pros and cons), and whether the two can work together.

---

### Official Review · Reviewer_X2hV · 2024-10-31

**Soundness:** 3
**Presentation:** 2
**Contribution:** 3
**Rating:** 6
**Confidence:** 4

**Summary:**

This paper introduces a distribution matching-based Best-of-N distillation method that simulates the Best-of-N distribution space, while reducing the time overhead of N inferences to just one. Starting from the theoretical distribution of BoN, the authors construct the Iterative BOND algorithm based on Quantile estimation and the choice of Jeffreys Divergence, and further propose the more practically meaningful J-BOND algorithm.

**Strengths:**

1. Rigorous Theoretical Analysis: This work rigorously analyzes the distribution characteristics under Best-of-N sampling and establishes its connection with standard RLHF, as well as the specific reward value $r_{BOND}$ under this correlation. This provides a reliable theoretical foundation for the work, rather than being based on naive assumptions.

2. Some Degree of Novelty: Although there is some concurrent work, the idea of distilling distributions from Best-of-N is fairly novel and important.

3. Consideration of Practical Efficiency: I appreciate the authors' consideration of the practical efficiency of the algorithm. The proposed J-BOND algorithm theoretically has lower sampling complexity, which should increase the efficiency of RLHF.

**Weaknesses:**

1. Lack of Important Baselines: Given that the main purpose of the paper is to distill Best-of-N sampling, BoN performance should straightforwardly serve as an important baseline to analyze pros and cons in terms of performance and efficiency. Moreover, other concurrent BoN distillation algorithms [1] should also be considered.

2. Lack of Downstream Validation: The main metrics in the paper, such as reward value and KL divergence, cannot be directly equated to the model's performance on downstream tasks. For an RLHF method, it is necessary to conduct experiments on downstream tasks and present more intuitive metrics to demonstrate the model's alignment performance.

3. Insufficient Experimental Setup: The paper lacks exploration of several issues. For instance, BoN sampling heavily depends on the Reward Model, and the influence of different RMs on the BOND algorithm is not investigated. Additionally, a more nuanced exploration of Jeffreys Divergence with smoother β variations could be included; and the comparison between J-BOND and standard Iterative BOND lacks investigation.

[1] Variational Best-of-N Alignment, https://arxiv.org/pdf/2407.06057

**Questions:**

1. Can the authors compare with more fundamental baseline methods, such as BoN or other BoN distillation algorithms?

2. Can the authors supplement additional experiments, including downstream validation and more ablation studies as discussed in Weakness 3?

3. Can the authors prove more clearly the advantages of the BOND algorithm over other Alignment algorithms, in terms of both performance and efficiency, to make the argument more convincing?

---

> ### Author Response · Authors · 2024-11-21
>
> Thank you for the positive feedback and constructive comments.
>
> Following your suggestions we have performed additional ablations and evaluations, as detailed in the general response.
> In particular, we have added:
> - Comparisons (both theoretical and experimental) with the concurrent work of Amini et al. (2024),
> - Additional ablations demonstrating the impact of Jeffreys divergence (as suggested, with smoother variations of $\beta$) in J-BOND,
> - Downstream evaluations of the Gemma 7B fine tuned policies, in terms of side-by-side comparisons and popular benchmark scores.
>
> We  refer to the general response for an overview on each of the above points.
>
> We hope these address the main concerns raised, but we are happy to expand them further if needed.

---

> > ### Comment · Reviewer_X2hV · 2024-11-24
> >
> > Thank you to the authors for the additional experiments and explanations; this has addressed most of my concerns.
> >
> > However, I am still curious about a few issues: How does the performance of BOND compare to actual Best-of-N sampling? How does BOND compare to other algorithms in terms of Reward Overoptimization? How is BOND's learning affected by the choice of Reward Model, and so on?
> >
> > Of course, I understand that addressing these issues might require more time. If some of these questions are resolved, I will consider raising the score.

---

> > > ### Author Response · Authors · 2024-11-25
> > >
> > > Thank you for being responsive and for the follow-up questions.
> > >
> > > The reason why we did not include Best-of-N sampling in our Gemma fine tuning experiments is simply because it would require a prohibitive large N to achieve comparable performance. Indeed, the considered RLHF baselines achieve best performance at a KL >> 50. The KL of BoN sampling grows logarithmically with N and thus – although BoN should achieve a better reward/KL tradeoff – it is impractical to run for achieving such high rewards.
> > > That said, because BoN sampling is the target distribution of BOND, we have compared against it in our XSum experiments. In Figure 7 (Appendix B.1) we compute actual KL distances between BOND policies and the BoN distribution. This can be done leveraging our analytical expression of Theorem 1 and using several MC samples to estimate quantiles. The decaying KL curves show that our BOND objective and its approximate gradients, are effective in terms of distilling the BoN distribution.
> > >
> > > In terms of reward overoptimization, we believe the Reward/KL tradeoff is quite a meaningful metric to compare different fine tuning algorithms: achieving better rewards without incurring a too high KL ensures the policy is not degenerating into exploitative behaviors of the reward model. The fact that J-BOND displays a better and stable reward/KL pareto curve is a good indication of reward hacking being mitigated. We believe this is quite interesting and encouraging, given that only log quantiles (i.e. generations’ rank) are optimized by the algorithm and not the actual rewards.
> > > The above considerations are ultimately validated by the end-to-end performance on side-by-side and zero-shot evals. J-BOND achieving the highest peak performance across the board is an effect of extracting the most out of the used reward model (note that reward hacking is unavoidable after a certain KL due to the reward model becoming out-of-distribution).
> > >
> > > Finally, BOND has absolutely minimal sensitivity to the used reward model (unlike other methods). This is because the RM is only used to compute ranks across generations, while its absolute values are unused. This makes J-BOND (and its hyperparameters) absolutely agnostic to whatever is the rewards’ signal range, shifts, and steepness.
> > >
> > > We hope the above answers resolve some of the remaining Reviewer’s concerns.

---

### Official Review · Reviewer_6VBe · 2024-11-02

**Soundness:** 4
**Presentation:** 4
**Contribution:** 3
**Rating:** 6
**Confidence:** 4

**Summary:**

The paper introduces Best-of-N Distillation (BOND), a novel alignment tuning algorithm designed to emulate the Best-of-N sampling method in a more computationally efficient manner. BOND aims to achieve the same high-quality output as Best-of-N sampling without the inference-time computational overhead, by aligning model outputs to match the distribution of the Best-of-N candidates.
In addition, to ensure stability and scalability, the authors introduce an iterative approximation strategy that operates effectively even with a minimal sample size (e.g., 2 or 4).
Further, based on the two types of loss function derivation aiming forward and reverse KL respectively, the author leverages Jeffreys divergence proposing J-BOND, an enhanced algorithm incorporating iterative distribution matching with an exponential moving average (EMA) anchor. J-BOND demonstrates effectiveness in maintaining a stable training and superior KL-reward trade-off through experiments.

**Strengths:**

The paper is well-structured and clearly presents its methodology, with detailed explanations and algorithms that allow readers to follow the progression. From iterative BOND to the addition of KL regularization in Sections 4 and 5, the additional experimental results effectively support these methodological advancements.
BOND is notable for its originality, offering a practical and computationally efficient alternative to traditional RLHF that achieves a superior KL-reward balance without requiring commitment to a specific regularization level. The work is significant in its potential impact on RLHF practices, as it provides a scalable solution for optimizing performance and efficiency while minimizing trade-offs between KL divergence from the reference distribution and reward maximization.

**Weaknesses:**

The paper relies heavily on the Jeffreys divergence without sufficient comparative analysis against alternative divergence metrics. The mode-covering and mode-seeking behavior property paper mentioned about are only observed in lower dimension such as multimodal distribution in 1-dimension. An inclusion of other divergence types, especially in the iterative stages, could offer clearer insights into the unique advantages of Jeffreys divergence. Further, relevant literature on divergence measures in alignment tuning, should also be cited to contextualize this choice.
- Go, Dongyoung, et al. "Aligning language models with preferences through f-divergence minimization." Proceedings of the 40th International Conference on Machine Learning. 2023.

As the paper discusses the method’s efficiency, the paper would benefit from explicit comparison of the computational cost saved by BOND relative to traditional Best-of-N sampling, or comparisons with sampling approaches used in RLHF. This would clarify BOND’s potential advantages in real-world applications.

Additionally, while the paper addresses the challenge of sampling size N through iterative approximation, showing practical advantages like non-saturation compared to non-iterative BOND, this helpful randomness raised in iterative BOND is solely introduced by approximation randomness, which lacks controls or specific directions. This calls into question whether the proposed algorithm genuinely achieves a distilled $\text{Bo}N^M$ distribution.
The substantial difference in $r(y)$ between iterative and non-iterative BOND  in Figure3 suggests a potential vulnerability to reward hacking, as discussed in Gao et al.
- Gao, Leo, John Schulman, and Jacob Hilton. "Scaling laws for reward model overoptimization." International Conference on Machine Learning. PMLR, 2023.


The introduction combines related works and problem setup, which could be structured more effectively. Detailed discussions on RLHF and Best-of-N would be more suitable in a separate related works section or could be incorporated into the problem setup. In the introduction, it would be clearer to emphasize the limitations of existing methods and highlight the advantages of the proposed approach over current methods.

**Questions:**

In Section 4.2, the authors utilize 32 Monte Carlo samples to estimate the backward and forward KL divergences between the training policy and the reference distribution. Given the high dimensionality of these distributions, this sample size seems insufficient to reliably capture the divergence and may introduce substantial estimation variance. A sensitivity analysis showing how the estimator's variance changes with an increasing number of Monte Carlo samples would strengthen the results. Alternatively, using a larger sample size for these estimates could enhance the reliability of the reported divergences.

While BOND’s benefits, such as improved KL/reward trade-offs and dynamic regularization, are discussed in the body of the paper, they are not clearly summarized in the introduction or abstract. A brief overview in these sections would effectively communicate BOND’s main advantages over traditional RLHF approaches, aiding readers in understanding its unique contributions and practical value.

---

> ### Author Response · Authors · 2024-11-21
>
> Thank you for your thorough review and positive feedback about our work. We would like to respond to the points raised.
>
>
> About the KL divergence estimates against the BoN distribution (Figure 7 of Appendix B.1), we agree that 32 MC samples introduce significant estimation variance. However, we remark that these are estimated with 32 *fresh* samples at each evaluation step. Each of such estimates correspond to a marker in Figure 2. As expected, there is quite some variance but we found the setup sufficiently informative to show clear trends and separations among the different baselines.
>
> To complement our results, we have performed additional ablations as detailed in the general response. In particular, in Appendix B.3 we have ablated the impact of different Jeffreys divergences in the (iterative) J-BOND algorithm. The ablation complement the one of Appendix B.1 demonstrating that mixing backward and forward KL is also beneficial to achieve better reward/KL tradeoffs.
> We thank the reviewer for providing us with the relevant work (Go et al. 2023) which we will cite in our manuscript. We acknowledge that alternative f-divergences can be used – we see this as an interesting future direction.
>
> Finally, we are happy to follow the Reviewer’s suggestions on making the benefits of our approach clearer in the introduction.

---

> > ### Comment · Reviewer_6VBe · 2024-11-26
> >
> > Thank you for your detailed response. I think to demonstrate the estimator's stability with 32 fresh samples, it would be best to present the corresponding MC simulation variation. The trends in Figure 3 (not Figure 2?) are not reliable unless simulated multiple times. However, your response does add clarity to the paper, and I have adjusted my score accordingly.

---

### Official Review · Reviewer_2AHH · 2024-11-05

**Soundness:** 3
**Presentation:** 3
**Contribution:** 3
**Rating:** 6
**Confidence:** 4

**Summary:**

The paper is focusing on the RLHF alignment problem, in particular on emulating the Best-of-N distribution which is known to perform very well, but is very costly at inference time (for each prompt it requires drawing N candidate generations from the reference model and selecting the one with highest reward according to a reward model). The authors propose the BOND (Best-of-N Distillation) algorithm designed to force the distribution of generations from the finetuned policy to be close to the Best-of-N distribution, requiring the generation of just a single sample (instead of N). To this end, BOND regards the alignment problem as a distribution matching problem and distills the Best-of-N distribution by finetuning the reference policy to imitate the Best-of-N distribution. To stay close to the original reference model, the authors incorporate a KL regularization term that considers both the forward and backward divergence (Jeffrey divergence). In addition, they incorporate Monte-Carlo quantile estimation, and exponential moving anchor, resulting in the J-BOND algorithm. The authors conduct experiments on the abstractive summarization task (XSum dataset) and aligning GEMMA using J-BOND.

**Strengths:**

The paper makes multiple contributions, namely theoretical derivation for the Best-of-N distribution and a practical RLHF finetuning algorithm that distills the Best-of-N distribution into a policy which is sample efficient and requires just one single sample at inference time

The authors are making a lot of engineering design choices in their proposed model, and carefully analyze the role of each component in the performance of the proposed algorithm

To regularize the model and ensure it is not steering too far from the reference model (the supervised finetuned policy), the authors use a combination of both forward and reverse KL, namely Jeffrey divergence. While the forward KL ensures mode covering behavior, the reverse KL is used for mode seeking behavior; their combination results in better aligned policies that combine the advantages of both divergences

Applying BOND recursively (Iterative BOND) improves the sample efficiency of the BOND algorithm and works for very small values of n (2, 4); its reward/KL tradeoff is comparable to the non-iterative BOND while being more sample efficient

The J-BOND algorithm presents better reward/KL trade-off compared to the REINFORCE algorithm with different values of \beta and does not require using a specific regularization strength

The paper is well written, well-motivated, presents theoretically and experimentally sound insights that would benefit the research community

**Weaknesses:**

The paper combines a lot of distinct ideas already proposed in previous works - it would be good to actually clearly articulate what the novel contribution is. Besides, the comparison with concurrent works is not very clear, in particular the difference with (Amini et al, 2024), WARM, WARP (Rame et al, 2024).

Figure 4 - It would be interesting to see how the performance of Best-of-N compares to the proposed algorithm J-BOND and REINFORCE

Algorithm 1, line 330 - \pi_t is not defined

Line 329 - \pi \in Capital \pi -Capital \pi  in Algorithm 1 is not defined

Line 456 - “a large previously trained reward model r(.)” - please provide details

Lines 481-482 - there are not many details about the hyperparameters of the REINFORCE algorithm

The authors are conducting experiments on Gemma 2B and 7B models, while results are convincing it would be good to see if they hold with other models and other tasks than summarization

**Questions:**

How does J-BOND performance compare to (Amini et al, 2024) and other concurrent works?

There are three algorithms discussed in the paper, namely BOND, Iterative-BOND and J-BOND. Is it always preferable to use JBOND or do you recommend using each algorithm in particular situations?

Will the code be made publicly available to serve the research community?

---

> ### Author Response · Authors · 2024-11-21
>
> We thank the Reviewer for the detailed review and the positive and constructive comments.
>
> We would like to clarify that BOND and iterative BOND are the two main building blocks of J-BOND, which is the ultimate algorithm we recommend for RLHF fine tuning. This is because BOND, as presented in Section 3 has the following main sources of complexity:
> 1. requires prescribing in advance to a fixed value of N
> 2. it does not scale with large N since the forward KL component requires sampling N times from the reference policy)
> 3. requires several per-prompt MC samples for accurate quantile estimation.
>
> J-BOND overcomes challenges (1) and (2) by employing the iterative BOND approach. In addition, it addresses challenge (3) by the crude quantile approximation based on 2 anchor samples.
>
>
> The challenges above can also serve to illustrate the differences between J-BOND and the concurrent vBoN approach by (Amini et al. 2024): vBoN does not suffer from (2), since it considers only backward KL divergence, but it crucially suffers from (1) and (3). In particular, (3) makes vBoN not viable for our fine-tuning experiments with conditional generations.
> As mentioned in the general response, we have clarified such points in the paper and performed additional ablations to compare J-BOND with a scalable version of vBoN with the same crude quantile estimation (see Appendix A.5 in the updated manuscript). Compared to vBoN, J-BOND does not require prescribing a fixed N in advance and displays a better reward/KL tradeoff. This is attributed to its iterative approach and to the additional forward KL minimization component.
>
> Finally, we thank the reviewer for the spotted inconsistencies – we will update the paper accordingly. We are planning to release the J-BOND code as well.

---

### Author Response · Authors · 2024-11-21
**General response**

We thank all the Reviewers for their valuable reviews and the positive feedback about our work.

While we are responding to each reviewer individually, we have followed common suggestions and complemented our paper with additional results.
In particular:
- We have compared J-BOND with the concurrent vBoN approach of (Amini et al. 2024), in Appendix A.5. First, we have highlighted the main differences between the two approaches and then compared them experimentally on Gemma 7B. In short, unlike J-BOND but similar to standard RLHF algorithms, vBoN requires fixing a regularization strength in advance (via the parameter $N$) and –moverover– considers only backward KL minimization which in (Amini et al. 2024) is estimated using several MC samples. Instead, J-BOND is designed to require only 2 anchor samples, optimizes Jeffreys divergence, and continuously optimize rewards achieving better reward/KL pareto solutions.

- We have performed additional ablations to demonstrate the impact of using different Jeffreys divergences in J-BOND. These are reported in Appendix B.3 and complement the Jeffreys divergence ablations of Appendix B.1. Utilizing Jeffreys divergence as objective, not only is beneficial to optimize both forward and backward KL divergences (as shown in Appendix B.1) but also enables J-BOND to achieve better reward/KL tradeoffs.

- In Appendix B.5, we have complemented our experiment with downstream evaluations of the Gemma 7B policies fine tuned by J-BOND and our standard RLHF (REINFORCE) baseline. More specifically, we reported side-by-side comparisons and zero-shot performance on several popular benchmarks. J-BOND achieves higher quality and significantly better scores across the board.

We are happy to provide further clarifications should there be any question.

---

### Meta-Review · Area_Chair_FY7g · 2024-12-18

**Metareview:**

This paper proposes to align an LM based on the best-of-N distribution. This paper is backed up by formal mathematical characterizations (which may be useful for others working with these types of distributions), as well as solid empirical results. There were some concerns with regard to justifying particular choices (e.g., Jeffrey's divergences), but I think this paper is a clear accept.

**Additional Comments On Reviewer Discussion:**

Several reviewers asked for comparison against vBoN (Amini et al. 2024). The authors, to their credit, performed comparison against vBoN during the rebuttal. However, these works are truly concurrent, and as such the comparison was not a factor in my decision.

---

### Decision · Program_Chairs · 2025-01-22

Accept (Poster)